# Not All Tasks are Equal - Task Attended Meta-learning for Few-shot Learning

## Abstract

Meta-learning (ML) has emerged as a promising direction in learning models under constrained resource settings like few-shot learning. The popular approaches for ML either learn a generalizable initial model or a generic parametric optimizer through batch episodic training. In this work, we study the importance of tasks in a batch for ML. We hypothesize that the common assumption in batch episodic training where each task in a batch has an equal contribution to learning an optimal meta-model need not be true. We propose to weight the tasks in a batch according to their "importance" in improving the meta-model's learning. To this end, we introduce a training curriculum called task attended meta-training to learn a meta-model from weighted tasks in a batch. The task attention module is a standalone unit and can be integrated with any batch episodic training regimen. Comparison of task-attended ML models with their non-task-attended counterparts on complex datasets, performance improvement of proposed curriculum over state-of-the-art task scheduling algorithms on noisy datasets, and cross-domain few shot learning setup validate its effectiveness.

## 1 Introduction

The ability to infer knowledge and discover complex representations from data has made deep learning models widely popular in the machine learning community. However, these models are data-hungry, often requiring large volumes of labeled data for training. Collection and annotation of such large amounts of training data may not be feasible for many real life applications, especially in domains that are inherently data constrained, like medical and satellite image classification, drug toxicity estimation, etc. Meta-learning (ML) has emerged as a promising direction for learning models in such settings, where only a limited amount (few-shots) of labeled training data is available. A typical ML algorithm employs an episodic training regimen that differs from the training procedure of conventional learning tasks. This episodic meta-training regimen is backed by the assumption that a machine learning model quickly generalizes to novel unseen data with minimal fine-tuning when trained and tested under similar circumstances (Vinyals et al., 2016). To facilitate such a generalization capacity, a meta-training phase is undertaken, where the model is trained to optimize its performance on several homogeneous tasks/episodes randomly sampled from a dataset. Each episode or task is a learning problem in itself. In the few-shot setting each task is a classification problem, a collection of $K$ support (train) and $Q$ query (test) samples corresponding to each of the $N$ classes. Task-specific knowledge is learned using the support data, and meta-knowledge across the tasks is learned using query samples, which essentially encodes "how to learn a new task effectively." The learned meta-knowledge is generic and agnostic to tasks from the same distribution. It is typically characterized in two different forms - either as an optimal initialization for the machine learning model or a learned parametric optimizer. Under the optimal initialization view, the learned meta-knowledge represents an optimal prior over the model parameters, that is equidistant, but close to the optimal parameters for all individual tasks. This enables the model to rapidly adapt to unseen tasks from the same distribution (Finn et al., 2017; Li et al., 2017; Jamal & Qi, 2019). Under the parametric optimizer view, meta-knowledge pertaining to the traversal of the loss surface of tasks is learned by the meta-optimizer. Through learning task specific and task agnostic characteristics of the loss surface, a parametric optimizer can thus effectively guide the base model to traverse the loss surface and achieve superior performance on unseen tasks from the same distribution (Ravi & Larochelle, 2017).

Initialization based ML approaches accumulate the meta-knowledge by simultaneously optimizing over a batch of tasks. On the other hand, a parametric optimizer sequentially accumulates meta-knowledge across individual tasks. The sequential accumulation process leads to a long oscillatory optimization trajectory and a bias towards the last task, limiting the parametric optimizer's task agnostic potential. However, recently meta-knowledge has been accumulated in a batch mode even for the parametric optimizer (Aimen et al., 2021). Further, under such batch episodic training (for both initialization and optimization views), a common assumption in ML that the randomly sampled episodes of a batch contribute equally to improving the learned meta-knowledge need not hold good. Due to the latent properties of the sampled tasks in a batch and the model configuration, some tasks may be better aligned with the optimal meta-knowledge than others. We hypothesize that proportioning the contribution of a task as per its alignment towards the optimal meta-knowledge can improve the meta-model's learning. This is analogous to classical machine learning algorithms like sample re-weighting, which however, operate at sample granularity. In re-weighting, samples leading to false positives are prioritized and therefore replayed. Hence, the latent properties due to which a sample is prioritized are explicitly defined. For complex task distributions, explicitly handcrafting the notion of "importance" of a task would be hard. To this end, we propose a task attended meta-training curriculum that employs an attention module that learns to assign weights to the tasks of a batch with experience. The attention module is parametrized as a neural network that takes meta-information in terms of the model's performance on the tasks in a batch as input and learns to associate weights to each of the tasks according to their contribution in improving the meta-model. Overall, we make the following contributions,

- We propose a task attended meta-training strategy wherein different tasks of a batch are weighted according to their "importance" defined by the attention module. This attention module is a standalone unit that can be integrated into any batch episodic training regimen.

- We extend the empirical investigation of the batch-mode parametric optimizer (MetaLSTM++) to complex datasets like miniImagenet, FC100, and tieredImagenet and validate its efficiency over its sequential counter-part (MetaLSTM).

- We conduct extensive experiments on miniImagenet, FC100, and tieredImagenet datasets and compare ML algorithms like MAML, MetaSGD, ANIL, and MetaLSTM++ with their task-attended counterparts to validate the effectiveness of the task attention module and its coupling with any batch episodic training regimen.

- We compare the proposed training curriculum with task-disagreement resolving approaches like TAML (Jamal & Qi, 2019) and conflict-averse gradient descent (Liu et al., 2021a) and validate the goodness of the proposed hypothesis. We extend these task-disagreement based approaches to the meta-learning regimen for a fair comparison.

- We further compare task-attended curriculum with state-of-the-art task scheduling approaches and also show the merit of the proposed approach on the miniImagenet-noisy dataset and cross-domain few shot learning (CDFSL) setup.

- We perform exhaustive empirical analysis and visual inspections to decipher the working of the task attention module.

## 2 Related Work

Transfer learning and meta-learning are two approaches that are commonly used to address few-shot learning problems. Transfer learning involves learning generalizable representations from larger datasets and models, and then using simple algorithms like fine-tuning to adapt to the specific task at hand. On the other hand, meta-learning approaches aim to find an algorithmic solution to few-shot learning. Due to their simplicity, transfer learning approaches scale well with larger image sizes and deeper models. In contrast, meta-learning approaches are memory intensive, which has become a barrier in scaling them to larger image sizes and deeper backbones (Dumoulin et al., 2021). Addressing the computational issues of meta-learning approaches and scaling them to larger support sets, deeper backbones and larger image sizes is a concurrent

area of research (Bronskill et al., 2021; Shin et al., 2021). We leave the integration of our approach with these techniques to enhance the scalability to the future. Equipped with deeper backbones and larger image sizes, transfer learning approaches achieved high performances, particularly in cross-domain settings (Bronskill et al., 2021; Guo et al., 2020; Dhillon et al., 2019; Dumoulin et al., 2021). However, a line of literature (Bronskill et al., 2021) suggests meta-learning approaches may be better suited for constrained test settings. This is because transfer learning relies on large pre-trained feature extractors and may require hundreds of optimization steps and careful hyperparameter tuning to perform well (Bronskill et al., 2021; Kolesnikov et al., 2020). For example, Meta-dataset Transfer approach (Triantafillou et al., 2019) finetunes all parameters of a ResNet18 feature backbone with a cosine classifier head for 200 optimization steps. Similarly, BiT (Kolesnikov et al., 2020) finetunes the feature backbone with a linear head, sometimes up to 20,000 optimization steps, to acquire state-of-the-art performance on VTAB dataset. Further, transfer learning approaches require significant hyper-parameter tuning on validation sets of each downstream task that also adds to the cost. On the other hand, meta-learning approaches can generalize to unseen meta-test tasks with just a few adaptation steps and often with little or no hyperparameter tuning (Bronskill et al., 2021). While transfer learning may be a better choice in some contexts, meta-learning can be a practical option in cases where computational resources are limited or when the task needs to be adapted on the fly. Overall, both approaches have their own strengths and can be useful in different settings. Our work focuses on a resource-constrained setting, where the number of support instances and the computing available for meta-test adaptation are limited. As a result, our study is confined to meta-learning setups.

ML literature is profoundly diverse and may broadly be classified into *initialization* (Finn et al., 2017; Li et al., 2017; Jamal & Qi, 2019; Raghu et al., 2020; Rusu et al., 2019; Sun et al., 2019) and *optimization approaches* (Ravi & Larochelle, 2017) depending on the metaknowledge. However, these approaches assume uniform contribution of tasks in learning a meta-model. In supervised learning, assigning non-uniform priorities to the samples is not new (Kahn & Marshall, 1953; Shrivastava et al., 2016). Self-paced learning (Kumar et al., 2010) and hard example mining (Shrivastava et al., 2016) have popularly been used to reweight the samples and various attributes like losses, gradients, and uncertainty have been used to assign priorities to samples (Lin et al., 2017; Zhao & Zhang, 2015; Chang et al., 2017). Zhao & Zhang (2015) introduce importance sampling to reduce variance and improve the convergence rate of stochastic optimization algorithms over uniform sampling. They theoretically prove that the reduction in the variance is possible if the sampling distribution depends on the norm of the gradients of the loss function. Chang et al. (2017) conclude that mini-batch SGD for classification is improved by emphasizing the uncertain examples. Lin et al. (2017) propose reshaped cross-entropy loss (focal loss) that down-weights the loss of confidently classified samples. Nevertheless, assigning non-uniform priorities to tasks in meta-learning is under-explored and has recently drawn attention (Kaddour et al., 2020; Gutierrez & Leonetti, 2020; Liu et al., 2020; Yao et al., 2021; Arnold et al., 2021). Gutierrez & Leonetti (2020) propose Information-Theoretic Task Selection (ITTS) algorithm to filter training tasks that are distinct from each other and close to the tasks of the target distribution. This algorithm results in a smaller pool of training tasks. A model trained on the smaller subset learns better than the one trained on the original set. On the other hand, Kaddour et al. (2020) propose probabilistic active meta-learning (PAML) that learns probabilistic task embeddings. Scores are assigned to these embeddings to select the next task presented to the model. These algorithms are, however, specific to meta-reinforcement learning (meta-RL). On the contrary, our focus is on the few shot classification problem. Liu et al. (2020) propose a greedy class-pair potential-based adaptive task sampling strategy wherein task selection depends on the difficulty of all class-pairs in a task. This sampling technique is static and operates at a class granularity. On the other hand, our approach is dynamic and operates at a task granularity. Assigning non-uniform weights to samples prevents overfitting on corrupt data points (Ren et al., 2018b; Jiang et al., 2018). Ren et al. (2018b) used gradient directions to re-weight the data points, and Jiang et al. (2018) learned a curriculum on examples using a mentor network. However, these approaches assume availability of abundant labeled data. Yao et al. (2021) extend Jiang et al. (2018) to the few-shot learning setup. They propose an adaptive task scheduler (ATS) to predict the sampling probability of tasks from a candidate pool containing a subset of tasks sampled from the original (noisy or imbalanced) task distribution (similar to (Jiang et al., 2018). Thus, the sampling probabilities of the tasks are (approximately) global. Another global task sampling approach is Uniform Sampling (Arnold et al., 2021), built on the premise that task difficulty (defined as the negative log-likelihood of the model on the task) approximately follows a normal

distribution and is transferred across model parameters during training. They also find sampling uniformly over episode difficulty outperforms other sampling schemes like curriculum, easy and hard mining. Our work is different from these approaches (ATS and Uniform Sampling) as we do not propose a global task sampling strategy but a dynamic task-batch re-weighting mechanism for the current meta-model update. We hypothesize that the task's importance depends on the data contained in it and the current meta-model's configuration. For example, in the initial stage of the meta-models training, coarse-grained tasks (tasks composed of semantically distinct classes) may have higher importance than fine-grained tasks (tasks composed of comparable classes), while this behavior may reverse as the training progresses. Further, our approach differs from Uniform Sampling in the definition of task difficulty, i.e., we neither explicitly handcraft the notion of task difficulty nor assume a normal distribution over it. Instead, we let an attention network learn the suitable weights for the tasks in a batch. Although ATS also dynamically learns the task sampling priority, it maintains a candidate pool to satisfy the global task priority criteria, causing overhead. Further, it performs an additional warm start to the scheduler, utilizes more task batches in a run, and uses REINFORCE for reward estimation; therefore, it is more expensive than the proposed approach. Contrary to our idea is TAML (Jamal & Qi, 2019) - a meta-training curriculum that enforces equity across the tasks in a batch. We show that weighting the tasks according to their "importance" and hence utilizing the diversity present in a batch given the meta-model's current configuration offers better performance than enforcing equity in a batch of tasks.

## 3 Preliminary

In a typical ML setting, the principal dataset $\mathcal{D}$ is divided into disjoint meta-sets $\mathcal{M}$ (meta-train set), $\mathcal{M}_v$ (meta-validation set) and $\mathcal{M}_t$ (meta-test set) for training the model, tuning its hyperparameters and evaluating its performance, respectively. Every meta-set is a collection of tasks $\mathcal{T}$ drawn from the joint task distribution $P(\mathcal{T})$ where each task $\mathcal{T}_i$ consists of support set $D_i = \{(x_k^c, y_k^c)_{k=1}^K\}_{c=1}^N$ and query set $D_i^* = \{(x_q^{*c}, y_q^{*c})_{q=1}^Q\}_{c=1}^N$. Here $(x, y)$ represents a (sample, label) pair and $N$ is the number of classes, $K$ and $Q$ are the number of samples belonging to each class in the support and query set, respectively. According to support-query characterization $\mathcal{M}$, $\mathcal{M}_v$ and $\mathcal{M}_t$ could be represented as $\{(D_i, D_i^*)\}_{i=1}^M$, $\{(D_i, D_i^*)\}_{i=1}^R$, $\{(D_i, D_i^*)\}_{i=1}^S$ where $M, R$ and $S$ are the total number of tasks in $\mathcal{M}$, $\mathcal{M}_v$ and $\mathcal{M}_t$ respectively. During meta-training, meta-model $\theta$ is adapted on $D_i$ of all tasks in a batch $\{\mathcal{T}_i\}_{i=1}^B$ of size $B$, $T$ times to obtain $\phi_i^T$. The adaptation occurs through gradient descent or parametric update on the train loss $L$ using learning rate $\alpha$. The adapted model $\phi_i^T$ is then evaluated on $D_i^*$ to obtain test loss $L^*$, which along with learning rate $\beta$, is used to update $\theta$. The output of this episodic training is either an optimal prior or a parametric optimizer, both aiming to facilitate the rapid adaptation of the model on unseen tasks from $\mathcal{M}_t$. The detailed note on initialization and optimization approaches is deferred to the supplementary material.

## 4 Task Attention in Meta-learning

A common assumption under the batch-wise episodic training regimen adopted by ML is that each task in a batch has an equal contribution in improving the learned meta-knowledge. However, this need not always be true. It is likely that given the current configuration of the meta-model, some tasks may be more important for the meta-model's learning. A contributing factor to this difference is that tasks sampled from complex data distributions can be profoundly diverse. The diversity and latent properties of the tasks coupled with the model configuration may induce some tasks to be better aligned with the optimal meta-knowledge than others. The challenging aspect in the meta-learning setting is to define the "importance" and associate weights to the tasks of a batch proportional to their contribution to improving the meta-knowledge. As human beings, we *learn* to associate importance to events subjective to meta-information about the events and prior experience. This motivates us to define a learnable module that can map the meta-information of tasks to their importance weights.

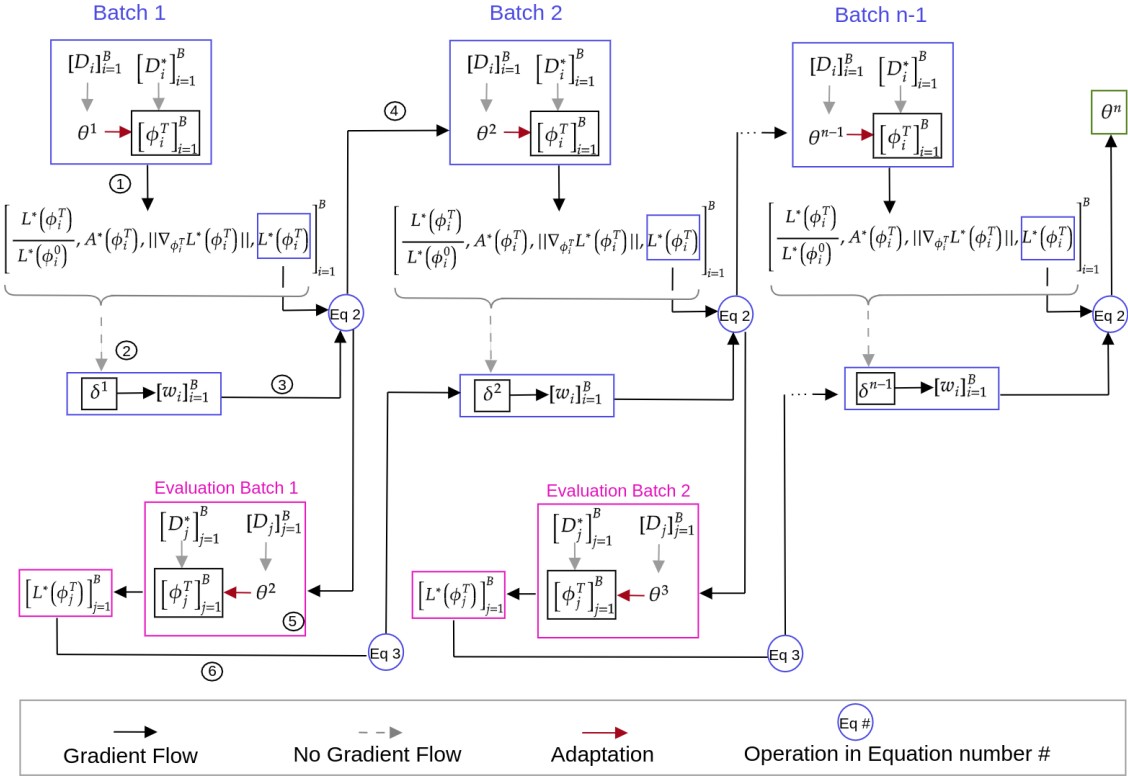

Figure 1: Computational Graph of the forward pass of the meta-model using task attended meta-training curriculum. The output of this procedure is a meta-model $\theta^n$. Gradients are propagated through solid lines and restricted through dashed lines.

### 4.1 Characteristics of Meta-Information

Given a task-batch $\{\mathcal{T}_i\}_{i=1}^B$, the task attention module takes as input meta-information about each task ($\mathcal{T}_i$) in the batch, defined as the four tuple below:

$$\mathcal{I} = \left\{ \left( ||\nabla_{\phi_i^T} L^*(\phi_i^T)||, L^*(\phi_i^T), A^*(\phi_i^T), \frac{L^*(\phi_i^T)}{L^*(\phi_i^0)} \right) \right\}_{i=1}^B \tag{1}$$

where corresponding to each task $i$ in the batch $||\nabla_{\phi_i^T} L^*(\phi_i^T)||$ denotes the norm of gradient, $L^*(\phi_i^T)$ and $A^*(\phi_i^T)$ are the test loss and accuracy of the adapted model respectively, and $\dfrac{L^*(\phi_i^T)}{L^*(\phi_i^0)}$ is the ratio of the model's test loss post and prior adaptation.

#### 4.1.1 Gradient Norm

Let $P = \left\{\phi_i^T\right\}_{i=1}^B$ be the parameters of the models obtained after adapting the initial model (for $T$ iterations) on the support data $\{D_i\}_{i=1}^B$ of tasks $\{\mathcal{T}_i\}_{i=1}^B$. Also, let $G = \left\{\nabla_{\phi_i^T} L^*(\phi_i^T)\right\}_{i=1}^B$ be the gradients of the adapted model parameters w.r.t the query losses $\{L^*(\phi_i^T)\}_{i=1}^B$. The gradient norm $\left\{||\nabla_{\phi_i^T} L^*(\phi_i^T)||\right\}_{i=1}^B$ is the $L_2$ norm of the gradients and quantifies the magnitude of the consolidated displacement of the adapted model parameters during a gradient descent update on query data. Larger gradient norm on query dataset could indicate that the model has either not learned the support set or has overfitted. Hence the model is not generalizable on query set compared to the models with low gradient norm. Gradient norm, therefore, carries information about the convergence and generalizability of the adapted models which has been theoretically studied in (Li et al., 2019).

**Algorithm 1:** Task Attended Meta-Training

**Input:**

*Dataset:* $\mathcal{M} = \{D_i, D_i^*\}_{i=1}^M$

*Models:* Meta-model $\theta$, Base-model $\phi$, Att-module $\delta$

*Learning-rates:* $\alpha$, $\beta$, $\gamma$

*Parameters:* Iterations $n_{iter}$, Batch-size $B$,
    Adaptation-steps $T$

**Output:** Meta-model $\theta$

**1 Initialization:** $\theta, \delta \leftarrow$ Random Initialization

**2 for** *iteration in $n_{iter}$* **do**

  **3**   $\{\mathcal{T}_i\}_{i=1}^B = \{D_i, D_i^*\}_{i=1}^B \leftarrow$ Sample task-batch$(\mathcal{M})$

  **4**   **for** *all $\mathcal{T}_i$* **do**

  **5**     $\phi_i^0 \leftarrow \theta$

  **6**     $L^*(\phi_i^0), \_ \leftarrow evaluate(\phi_i^0, D_i^*)$    $\triangleright$ Compute loss
       and accuracy of input model on given dataset.

  **7**     $\phi_i^T = adapt(\phi_i^0, D_i)$

  **8**     $L^*(\phi_i^T), A^*(\phi_i^T) \leftarrow evaluate(\phi_i^T, D_i^*)$

  **9**   **end**

  **10**   $[w_i]_{i=1}^B \leftarrow Att\_module$

$$\left( \left[ \frac{L^*(\phi_i^T)}{L^*(\phi_i^0)}, A^*(\phi_i^T), ||\nabla_{\phi_i^T} L^*(\phi_i^T)||, L^*(\phi_i^T) \right]_{i=1}^B \right)$$

  **11**   $\theta \leftarrow \theta - \beta\nabla_\theta \sum_{i=1}^B w_i L^*(\phi_i^T)$

  **12**   $\{D_j, D_j^*\}_{j=1}^B \leftarrow$ Sample task-batch$(\mathcal{M})$

  **13**   **for** *all $\mathcal{T}_j$* **do**

  **14**     $\phi_j^0 \leftarrow \theta$

  **15**     $\phi_j^T = adapt(\phi_j^0, D_j)$

  **16**   **end**

  **17**   $\delta \leftarrow \delta - \gamma\nabla_\delta \sum_{j=1}^B L^*(\phi_j^T)$

**18 end**

**19 Return** $\theta$

**20 Function** adapt$(\phi_i^t, D_i)$**:**

  **21**   $\theta \leftarrow \phi_i^t$

  **22**   **if** *$\theta$ is optimal-initialization* **then**

  **23**     **for** *t=1 to T* **do**

  **24**       $\phi_i^{t+1} \leftarrow \phi_i^t - \alpha\nabla_{\phi_i^t} L(\phi_i^t)$

  **25**     **end**

  **26**   **end**

  **27**   **else if** *$\theta$ is parametric-optimizer* **then**

  **28**     **for** *t=1 to T* **do**

  **29**       $\phi_i^{t+1} \leftarrow \theta\left(L(\phi_i^t), \nabla_{\phi_i^t} L(\phi_i^t)\right)$    $\triangleright$ Parameter
        updates given by cell state of $\theta$.

  **30**     **end**

  **31**   **end**

**32 Return** $\phi_i^T$

### 4.1.2 Test Loss

$\{L^*(\phi_i^T)\}_{i=1}^B$ represents the empirical error (cross entropy loss) of the adapted base models on unseen query instances and hence characterizes their generalizability. Unlike gradient norm, which characterizes the generalizability in parameter space, query loss quantifies generalizability in the output space as the divergence between the real and predicted probability distributions. As $\{L^*(\phi_i^T)\}_{i=1}^B$ is a key component in the meta-update equation, it is an important factor influencing the meta-model's learning. Further, test errors of classes have been widely used to determine their "easy or hardness" (Bengio et al., 2009; Liu et al., 2021b; Arnold et al., 2021). Thus $\{L^*(\phi_i^T)\}_{i=1}^B$ acquaints the attention module with the generalizability aspect of task models and their influence in updating the meta-model.

### 4.1.3 Test Accuracy

$\{A^*(\phi_i^T)\}_{i=1}^B$ corresponds to the accuracies of $\{\phi_i^T\}_{i=1}^B$ on $\{D_i^*\}_{i=1}^B$ scaled in the range [0,1]. $A^*(\phi_i^T)$ evaluates the thresholded predictions (predicted labels) unlike $L^*(\phi_i^T)$, which evaluates the confidence of the model's predictions on the true class labels. Two task models may predict the same class labels but differ in the confidence of the predictions. In such scenarios, neither loss nor accuracy is individually sufficient to comprehend this relationship among the tasks. So, the combination of these two entities is more reflective of the nature of the learned task models.

### 4.1.4 Loss-ratio

Let $L^*(\phi_i^0)$ be the loss of $\theta$ on the $D_i^*$, and $L^*(\phi_i^T)$ be the loss of the adapted model $\phi_i^T$ on $D_i^*$. The loss-ratio $\frac{L^*(\phi_i^T)}{L^*(\phi_i^0)}$ is representative of the relative progress of a meta-model on each task. Higher values ($> 1$) of the loss-ratio suggests adapting $\theta$ to $D_i$ has an adverse effect on generalizing it to $D_i^*$ (negative impact), while lower values ($< 1$) of the loss-ratio indicates the benefit of adaptation of $\theta$ on $D_i$ (positive impact). Loss-ratio of exactly one signifies adaptation attributes to no additional benefit (neutral impact). Therefore, loss-ratio provides information regarding the impact of adaptation on each task for a given meta-model.

## 4.2 Task Attention Module

We learn a task attention module parameterized by $\delta$, which attends to the tasks that contribute more to the model's learning i.e., the objective of the task attention module is to learn the relative importance of

each task in the batch for the meta-model's learning. Thus the output of the module is a $B-$dimensional vector $\mathbf{w} = [w_1, \ldots, w_B]$, ($\sum_{i=1}^{B} w_i = 1$ and $\forall \mathcal{T}_i$, $w_i \geq 0$) quantifying the attention-score (weight - $w_i$) for each task. The attention vector $\mathbf{w}$ is multiplied with the corresponding task losses of the adapted models $L^*(\phi_i^T)$ on the held-out datasets $D_i^*$ to update the meta-model $\theta$:

$$\theta^{t+1} \leftarrow \theta^t - \beta \nabla_{\theta^t} \sum_{i=1}^{B} w_i L^*(\phi_i^T) \tag{2}$$

After the meta-model is updated using the weighted task losses, we evaluate the goodness of the generated attention weights. We sample a new batch of tasks $\{D_j, D_j^*\}_{j=1}^{B}$ and adapt a base-model $\phi_j$ using the updated meta-model $\theta^{t+1}$ on the train data $\{D_j\}$ of each task. The mean test-loss of the adapted models $\{\phi_j^T\}_{j=1}^{B}$ reflect the goodness of the weights assigned by the attention-module in the previous iteration. The attention module $\delta$ is thus updated using the gradients flowing back into it w.r.t to this mean test-loss. The attention network is trained simultaneously with the meta-model in an end to end fashion using the update rule:

$$\delta^{t+1} \leftarrow \delta^t - \gamma \nabla_{\delta^t} \sum_{j=1}^{B} L^*(\phi_j^T) \tag{3}$$

where $\phi_j^T$ is adapted from $\theta^{t+1}$ and $\gamma$ is the learning rate.

## 4.3  Task Attended Meta-Training Algorithm

We demonstrate the meta-training curriculum using the proposed task attention in Figure 1 and formally summarize it in Algorithm 1. The detailed explanation is presented in Figure 7 in the appendix. As with the classical meta-training process, we first sample a batch of tasks from the task distribution. For each task $\mathcal{T}_i$, we adapt the base-model $\phi_i$ using the train data $D_i$ for $T$ time-steps (line 7 and lines 20-32 in Algorithm 1). Specifically, for initialization approaches, adaptation is performed by gradient descent on train loss $L$ (lines 22-26 in Algorithm 1). However, for optimization approaches, current loss and gradients are inputted to the meta-model $\theta$, which outputs the updated base-model parameters (lines 27-31 in Algorithm 1). Then we compute the meta-information about the adapted model corresponding to each task. It comprises of the loss $L^*(\phi_i^T)$, accuracy $A^*(\phi_i^T)$, loss-ratio $\dfrac{L^*(\phi_i^T)}{L^*(\phi_i^0)}$ and gradient norm $||\nabla_{\phi_i^T} L^*(\phi_i^T)||$ on the test data $D_i^*$. This meta-information corresponding to each task in a batch is given as input to the task attention module (Figure 1 - Label: ②) which outputs the attention vector (line 10 in Algorithm 1). The attention vector and test losses $\{L^*(\phi_i^T)\}_{i=1}^{B}$ are used to update meta-model parameters $\theta$ according to equation 2 (line 11 in Algorithm 1, Figure 1 - Label: ④). We sample a new batch of tasks $\{D_j, D_j^*\}_{j=1}^{B}$ and adapt the base-models $\{\phi_j^T\}_{j=1}^{B}$ using the updated meta-model (lines 12-16 in Algorithm 1, Figure 1 - Label: ⑤). We compute the mean test loss over the adapted base-models $\{L^*(\phi_j^T)\}_{j=1}^{B}$, which is then used to update the parameters of the task attention module $\delta$ according to equation 3 (line 17 in Algorithm 1, Figure 1 - Label: ⑥).

The attention network is designed as a stand-alone module to learn the mapping from the meta-information space to the importance of tasks in a batch. The meta-model is learned according to equation 2 and aims to minimize the weighted loss. It is important to decouple the learning of the attention network from that of the meta-model. If there is information flow from the task attention module to the meta-model, the latter may reduce its weighted loss by learning an initialization that is suboptimal, but for which the task attention network assigns lower weights. This would introduce an undesirable bias to the learning process. To circumvent this bias, we restrict the flow of gradients to the meta-model $\theta$ through the task attention module $\delta$ by enforcing $\nabla_\theta w_i L^*(\phi_i^T) = w_i \nabla_\theta L^*(\phi_i^T)$ i.e., $\nabla_\theta w_i$ is not computed. Also, gradients flowing through the attention network to the meta-model create additional computational overhead. Specifically, the term $\nabla_\theta \sum_i w_i L^*(\phi_i^T)$ from equation 2 can be expanded as follows -

$$\nabla_\theta \sum_i w_i L^*(\phi_i^T) = \sum_i \nabla_\theta w_i L^*(\phi_i^T) = \underbrace{\sum_i w_i \nabla_\theta L^*(\phi_i^T)}_{\text{Term 1}} + \underbrace{\sum_i L^*(\phi_i^T) \nabla_\theta w_i}_{\text{Term 2}}$$

278  The $\nabla_\theta w_i$ in Term 2 is computationally expensive as $\nabla_\theta w_i = \nabla_\delta w_i . \nabla_I \delta . \nabla_\phi I . \nabla_\theta \phi$. Restricting the gradient
279  flow avoids these additional computations. We also note that the meta-model and attention network are
280  updated only once during each training iteration, although on different batches of tasks.

## 5  Experiments and Results

We conduct experiments to demonstrate the merit of the task-attention across multiple datasets, training
setups, and learning paradigms. We verify that the proposed regimen could be integrated with various
ML approaches like MAML, MetaSGD, MetaLSTM++, and ANIL and further show its superiority over
state-of-the-art task-scheduling and conflict-resolving approaches. We also analyze the attention network.

### 5.1  Dataset and Implementation Details

In line with the state-of-the-art literature (Sun et al., 2020; Arnold et al., 2021), we use miniImagenet, FC100, and tieredImagenet for evaluating the effectiveness of the proposed attention module as they are more challenging datasets comprising of highly diverse tasks. We also test the efficacy of the proposed approach on noisy dataset (miniImagenet-noisy), and under cross-domain few shot learning (CDFSL) miniImagenet $\rightarrow$ CUB-200 and mini-Imagenet $\rightarrow$ FGVC-Aircrafts datasets. The details of the datasets are presented in the supplementary material.

We use a 4-layer CNN from (Finn et al., 2017) as a base model and a two-layer LSTM (Ravi & Larochelle, 2017) for the parametric optimizer. The architecture of the task-attention module is illustrated in Figure 2 and de-

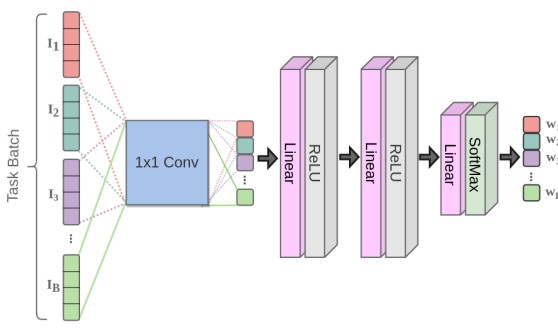

Figure 2: Architecture of Task-attention module.

scribed as follows.The task attention module is implemented as a 4-layer neural network. The first layer performs a 1×1 convolution over the input (meta-information) of size B×4 where B denotes the meta-batch size, producing a vector of size B×1 as output. This vector is then passed through two fully connected layers with 32 hidden nodes, each followed by a ReLU activation. This output is then passed through a fully connected layer with B nodes, followed by a softmax activation to produce the normalized attention weights.

We perform a grid search over 30 different configurations for 5000 iterations to find the optimal hyper-parameters for each setting. The search space is shared across all meta-training algorithms and datasets. The meta, base and attention model learning rates are sampled from a log uniform distribution in the ranges $[1e^{-4}, 1e^{-2}]$, $[1e^{-2}, 5e^{-1}]$ and $[1e^{-4}, 1e^{-2}]$ respectively (see appendix for more details). The hyperparameter $\lambda$ for TAML (Theil) is sampled from a log uniform distribution over the range of $[1e^{-2}, 1]$. For CA-MAML, c is set as 0.5. The meta-batch size is set to 4 for all settings (Finn et al., 2017; Jamal & Qi, 2019).

Table 1: Comparison of few-shot classification performance of MAML and TA-MAML on miniImagenet dataset with meta-batch size 4 and 6 and 8 for 5 and 10 way (1 and 5 shot) settings. The $\pm$ represents the 95% confidence intervals over 300 tasks. Algorithms denoted by * are rerun on their optimal hyper-parameters on our experimental setup. We observe that TA-MAML consistently performs better than MAML, and an increase in the tasks in a batch improves the performance of both MAML and TA-MAML.

| | Test Accuracy (%) on miniImagenet | | | |
| --- | --- | --- | --- | --- |
| | 5 Way | | 10 Way | |
| **Model** | 1 Shot | 5 Shot | 1 Shot | 5 Shot |
| | **Batch Size 4** | | | |
| MAML* | 46.10 ± 0.19 | 60.16 ± 0.17 | 29.42 ± 0.11 | 41.98 ± 0.10 |
| **TA-MAML*** | **48.36 ± 0.23** | **62.48 ± 0.18** | **31.15± 0.11** | **43.70 ± 0.09** |
| | **Batch Size 6** | | | |
| MAML* | 47.72 ± 1.041 | 63.45 ± 1.083 | 31.55 ± 0.626 | 46.27 ± 0.64 |
| **TA-MAML*** | **49.14 ± 1.211** | **65.26 ± 0.956** | **32.62± 0.635** | **46.67 ± 0.63** |
| | **Batch Size 8** | | | |
| MAML* | 47.68±1.20 | 63.81±0.98 | 31.54±0.66 | 46.15±0.58 |
| **TA-MAML*** | **50.35±1.22** | **65.69±1.08** | **32.00±0.68** | **48.33±0.63** |

However, we study its impact in Table 1. All models were trained for 55000 iterations (early stopping was employed for tieredImagenet) using the optimal set of hyper-parameters using an Adam optimizer (Kingma & Ba, 2015). All the experimental results and comparisons correspond to our re-implementation of the ML algorithms integrated into learn2learn library (Arnold et al., 2020) to ensure fairness and uniformity. We believe that integrating the proposed attention module and additional ML algorithms into the learn2learn library will benefit the ML community. We perform individual hyperparameter tuning for all the models over the same hyperparameter space to ensure a fair comparison. The source code is publicly available.[1]

The literature reports significant variations in the meta-test performances of various ML approaches (Table 7 in supplementary material). The reported average meta-test accuracies of MAML on the miniImagenet dataset range from 46.47 % to 48.70 % (55.16% to 64.39%) for 5 way 1 shot (5 shot) settings. A careful analysis reveals the different experimental setups resulting in the observed variation. Experimental setups (Finn et al., 2017; Oreshkin et al., 2018; Oh et al., 2020) differ in the number of examples per class in the query set, the number of gradient descent steps in the inner loop, meta-batch size, inductive or transductive batch normalization, etc. We conduct two sets of experiments to test the proposed task attention model's efficacy in a fair manner. The first set of experiments use the train and test setups reported in the literature (denoted using $^{\#}$). The second set uses our setup (denoted using $^{*}$) that has the same train and test conditions. Specifically, we set the query examples per class to 15 and gradient steps to 5 for both the meta-train and meta-test phases. However, for 10 way 5 shot setting, we use only 2 gradient steps to reduce the computational burden. More query examples per class (15) during the meta-test provide a robust estimate of the model's generalizability. Further, setting gradient steps to 5 (or 2) better evaluates the quick adaptation capabilities of a learned prior.

## 5.2 Influence of Task Attention on Meta-Training

As task-attention (TA) is a standalone module, it can be integrated with any batch episodic training regimen. We, therefore, use MetaLSTM++ (batch mode of MetaLSTM) for our experiments. In (Aimen et al., 2021), authors demonstrated the merit of MetaLSTM++ on MetaLSTM only on Omniglot dataset. We extend upon this empirical investigation by comparing the performance of MetaLSTM and MetaLSTM++ on complex datasets like miniImagenet, FC100, and tieredImagenet (Table 2). It is evident from the results that batch-wise episodic training is more effective than sequential episodic training.

We also investigate the performance of models trained with the TA meta-training regimen with their non-TA counterparts on both (our and reported - wherever available) setups. Specifically, we compare MAML, MetaSGD, MetaLSTM++, and ANIL with their task-attended versions on 5 and 10 way (1 and 5 shot) settings on mini-Imagenet, FC100, and tieredImagenet datasets and report the results in Table 2. We consider 300 meta-test tasks for all approaches unless specified otherwise. For

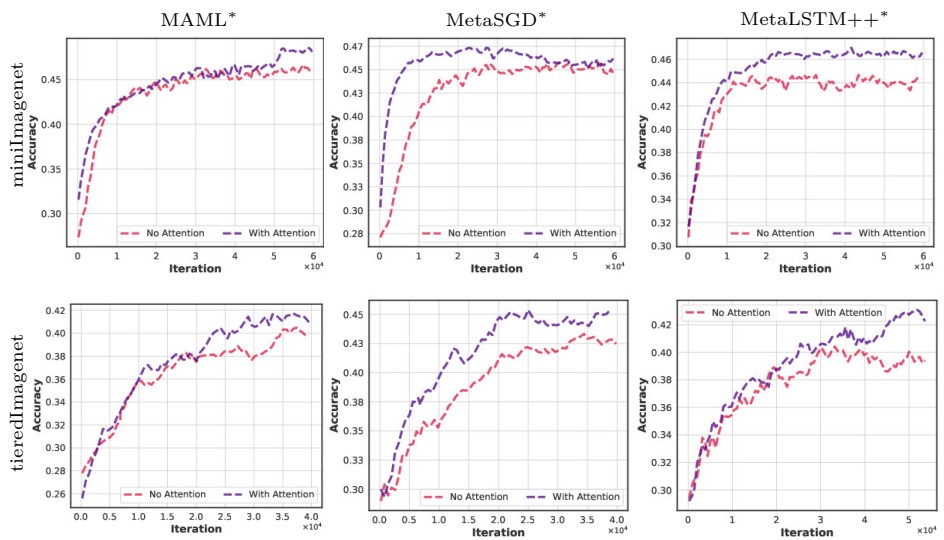

Figure 3: Mean validation accuracies of MAML$^{*}$ (Col-1), MetaSGD$^{*}$ (Col-2) and MetaLSTM++$^{*}$ (Col-3) across 300 tasks with/without attention on 5 way 1 shot setting on miniImagenet (Row-1) and tieredImagenet (Row-2) datasets.

[1]https://github.com/taskattention/task-attended-metalearning.git

Table 2: Comparison of few-shot classification performance of vanilla ML algorithms with their task attended versions on miniImagenet, FC100 and tieredImagenet datasets for 5 and 10 way (1 and 5 shot) settings. The $\pm$ represents the 95% confidence intervals over 300 tasks. Algorithms denoted by * and # are rerun on using the optimal hyper-parameters on our and reported experimental setups, respectively. Attention-based ML algorithms perform better than their corresponding vanilla approaches across all the settings. Further, MetaLSTM++ and TA-MAML perform better than MetaLSTM and TAML (and CA-MAML) , respectively, across all settings and datasets.

| | Test Accuracy (%) | | | |
| | 5 Way | | 10 Way | |
| **Model** | 1 Shot | 5 Shot | 1 Shot | 5 Shot |
|---|---|---|---|---|
| | **miniImagenet** | | | |
| MAML[#](Finn et al., 2017) | 48.07 $\pm$ 1.75 | 63.15 $\pm$ 0.91 | - | - |
| CA-MAML[#](Liu et al., 2021a) | 47.86 $\pm$ 2.50 | 64.27 $\pm$ 1.26 | - | - |
| TAML[#](Jamal & Qi, 2019) | 51.77 $\pm$ 1.86 | 65.6 $\pm$ 0.93 | - | - |
| **TA-MAML[#]** | **53.80 $\pm$ 1.85** | **66.11 $\pm$ 0.11** | - | - |
| MAML[*] | 46.10 $\pm$ 0.19 | 60.16 $\pm$ 0.17 | 29.42 $\pm$ 0.11 | 41.98 $\pm$ 0.10 |
| TAML[*] | 46.26 $\pm$ 0.21 | 53.40 $\pm$ 0.14 | 29.76 $\pm$ 0.11 | 36.88 $\pm$ 0.10 |
| **TA-MAML[*]** | **48.36 $\pm$ 0.23** | **62.48 $\pm$ 0.18** | **31.15$\pm$ 0.11** | **43.70 $\pm$ 0.09** |
| MetaSGD[#] (Li et al., 2017) | 50.47 $\pm$ 1.87 | 64.03 $\pm$ 0.94 | - | - |
| **TA-MetaSGD[#]** | **52.60 $\pm$ 0.25** | **67.54 $\pm$ 0.12** | - | - |
| MetaSGD[*] | 47.65$\pm$ 0.21 | 61.60 $\pm$ 0.17 | 30.09$\pm$ 0.10 | 42.22 $\pm$ 0.11 |
| **TA-MetaSGD[*]** | **49.28 $\pm$ 0.20** | **63.37 $\pm$ 0.16** | **31.50$\pm$ 0.11** | **44.06 $\pm$ 0.10** |
| MetaLSTM[*] | 41.48 $\pm$ 1.02 | 58.87 $\pm$ 0.94 | 28.62 $\pm$ 0.64 | 44.03 $\pm$ 0.69 |
| MetaLSTM++[*] | 48.00 $\pm$ 0.19 | 62.73 $\pm$ 0.17 | 31.16 $\pm$ 0.09 | 45.46 $\pm$ 0.10 |
| **TA-MetaLSTM++[*]** | **49.18 $\pm$ 0.17** | **64.89 $\pm$ 0.16** | **32.07$\pm$ 0.11** | **46.66 $\pm$ 0.09** |
| ANIL[#](Raghu et al., 2020) | 46.7 $\pm$ 0.4 | 61.5 $\pm$ 0.5 | - | - |
| **TA-ANIL[#]** | **49.53 $\pm$ 0.41** | **63.73 $\pm$ 0.33** | - | - |
| ANIL[*] | 46.92 $\pm$ 0.62 | 58.68 $\pm$ 0.54 | 28.84 $\pm$ 0.34 | 40.95 $\pm$ 0.32 |
| **TA-ANIL[*]** | **48.84 $\pm$ 0.62** | 60.80$\pm$ 0.55 | **31.14$\pm$ 0.34** | **42.52 $\pm$ 0.34** |
| | **FC100** | | | |
| MAML[*] | 36.40 $\pm$ 0.38 | 46.76$\pm$0.21 | 23.93$\pm$0.14 | 31.14 $\pm$ 0.07 |
| TAML[*] | 38.00 $\pm$ 0.26 | 48.05$\pm$ 0.13 | 21.60$\pm$ 0.14 | 33.19$\pm$ 0.07 |
| **TA-MAML[*]** | **39.86$\pm$ 0.25** | **49.56 $\pm$ 0.13** | **25.46$\pm$ 0.15** | **36.06$\pm$ 0.08** |
| MetaSGD[*] | 33.46 $\pm$ 0.23 | 43.96$\pm$ 0.13 | 21.40$\pm$0.15 | 30.59$\pm$ 0.07 |
| **TA-MetaSGD[*]** | **35.66$\pm$0.25** | **49.49$\pm$ 0.12** | **23.80$\pm$0.15** | **32.08$\pm$0.07** |
| MetaLSTM[*] | 37.20 $\pm$ 0.26 | 47.89 $\pm$ 0.13 | 21.70 $\pm$ 0.14 | 32.11 $\pm$ 0.07 |
| MetaLSTM++[*] | 38.60 $\pm$0.23 | 49.82 $\pm$ 0.12 | 22.80 $\pm$ 0.14 | 33.46 $\pm$ 0.08 |
| **TA-MetaLSTM++[*]** | **41.53 $\pm$0.28** | **51.17 $\pm$0.13** | **25.33 $\pm$0.15** | **34.18 $\pm$0.08** |
| ANIL[*] | 34.08 $\pm$ 1.29 | 44.74 $\pm$ 0.68 | 20.65 $\pm$ 0.77 | 27.93 $\pm$ 0.42 |
| **TA-ANIL[*]** | **38.06 $\pm$ 1.26** | **46.94$\pm$ 0.69** | **23.27$\pm$ 0.79** | **28.29 $\pm$ 0.40** |
| | **tieredImagenet** | | | |
| MAML[#](Oh et al., 2020) | 47.44 $\pm$ 0.18 | 64.70 $\pm$ 0.14 | - | - |
| **TA-MAML[#]** | **51.90 $\pm$ 0.19** | **69.43$\pm$ 0.18** | - | - |
| MAML[*] | 44.40 $\pm$ 0.49 | 57.07 $\pm$ 0.22 | 27.40 $\pm$ 0.25 | 34.30 $\pm$ 0.14 |
| TAML[*] | 46.40 $\pm$ 0.40 | 56.80 $\pm$ 0.23 | 26.40 $\pm$ 0.25 | 34.40 $\pm$ 0.15 |
| **TA-MAML[*]** | **48.40 $\pm$ 0.46** | **60.40 $\pm$ 0.25** | **31.00$\pm$ 0.26** | **37.60$\pm$ 0.15** |
| MetaSGD[*] | 52.80 $\pm$ 0.44 | 62.35 $\pm$ 0.26 | 31.90 $\pm$ 0.27 | 44.16 $\pm$ 0.15 |
| **TA-MetaSGD[*]** | **56.20 $\pm$ 0.45** | **64.56 $\pm$ 0.24** | **33.20$\pm$ 0.29** | **47.12 $\pm$ 0.16** |
| MetaLSTM[*] | 37.00 $\pm$ 0.44 | 59.83 $\pm$ 0.25 | 29.80 $\pm$ 0.28 | 39.28 $\pm$ 0.13 |
| MetaLSTM++[*] | 47.60 $\pm$ 0.49 | 63.24 $\pm$ 0.25 | 30.70 $\pm$ 0.27 | 47.97 $\pm$ 0.16 |
| **TA-MetaLSTM++[*]** | **49.00 $\pm$ 0.44** | **66.15 $\pm$ 0.23** | **32.10$\pm$ 0.27** | **51.35 $\pm$ 0.17** |
| ANIL[*] | 45.08 $\pm$ 1.37 | 59.71 $\pm$0.77 | 29.32 $\pm$ 0.83 | 42.76 $\pm$ 0.50 |
| **TA-ANIL[*]** | **45.96 $\pm$ 1.32** | **60.96$\pm$ 0.72** | **32.68$\pm$ 0.92** | **47.56 $\pm$ 0.51** |

377 ANIL and its task-attended counterpart, we consider 1000 testing tasks. From Table 2, we observe that
378 models trained with TA regimen generalize better to the unseen meta-test tasks than their non-task-attended
379 versions across all the settings in all datasets. Note that the proposed task attention mechanism aims not
380 to surpass the state-of-the-art meta-learning algorithms but provides new insight into the batch episodic
381 meta-training regimen, which as per our knowledge, is common to all meta-learning algorithms.

382 We also compare the performance of TA-MAML against TAML - a meta-training regimen that forces the
383 meta-model to be equally close to all the tasks. The results, as presented in Table 2, suggest that TA-MAML
384 performs better than TAML on all benchmarks across all settings. Note that both TAML and TA-MAML
385 are approaches that built upon MAML to address the inequality/diversity of tasks in a batch. Our aim is
386 thus to compare TAML and TA-MAML and not to assess the efficacy of TAML when meta-trained using
387 task attention. Liu et al. (2021a) proposed an optimization method to neutralize conflicts of an average
388 model with individual tasks in a multi-task learning setup. Specifically, they find an optimal update vector
389 that lies in the proximity of the average gradient across the batch of the tasks without conflicting with any
390 task gradient. This method is similar to (Jamal & Qi, 2019) in a meta-learning setup, which constrains
391 the losses of tasks towards the average loss on a task batch. As the update vector is constrained to be
392 close to the average gradient vector on a task batch, information flow from certain useful tasks to the meta-
393 model may decrease. We note that we extend (Liu et al., 2021a) to a meta-learning setup by computing
394 the average and weighted average gradients on query loss of the adapted models instead of a model from
395 the previous iteration (as in a multi-task setup). Table 2 demonstrates that the proposed attention mech-
396 anism has better generalizability to unseen tasks than conflict-averse gradient descent adapted for a meta-
397 learning setup (CA-MAML). Our approach utilizes a non-linear model to extract knowledge from multiple
398 meta-information components to learn the weights, which helps it to outperform TAML and CA-MAML.

399

400 We investigate the influence of the TA meta-
401 training regimen on the model's convergence by
402 analyzing the trend of the model's validation ac-
403 curacy over iterations. Figure 3 depicts the mean
404 validation accuracy over 300 tasks on miniImagenet
405 and tieredImagenet datasets for a 5 way 1 shot set-
406 ting across training iterations. We observe that
407 the models meta-trained with TA regimen tend to
408 achieve higher/at-par performance in fewer itera-
409 tions than the corresponding models meta-trained
410 with the non-TA regimen.

### 5.3 Comparison with Sampling Approaches

412 We compare our proposed approach with ATS (Yao
413 et al., 2021) and uniform sampling (Arnold et al.,
414 2021) and demonstrate that our weighting mecha-
415 nism imparts better generalizability to the meta-
416 model than the global weighting of the tasks.
417 Yao et al. (2021) ascertained the merit of ATS
418 over Greedy class-pair (GCP) technique (Liu et al.,
419 2020) on miniImagenet dataset. We extend this
420 comparison and show in Table 3 that the pro-
421 posed approach performs better than state-of-the-
422 art ATS and GCP in both 1 and 5 shot settings.
423 We also observe that the TA mechanism performs
424 better than uniform sampling on the miniImagenet
425 dataset on 1 and 5 shot settings for MAML and
426 ANIL. ATS has been designed for noisy and im-
427 balanced task distributions. So, we compare the

Table 3: Comparison (Test Accuracy (%)) of task attention with GCP, ATS and Uniform Sampling for MAML and MetaSGD (or ANIL) on miniImagenet dataset and various sampling techniques for ANIL on the miniImagenet-noisy dataset for 5 way 1 and 5 shot settings. For miniImagenet, algorithms denoted by * and # are rerun on the optimal hyper-parameters on our and reported experimental setups, respectively.

| Model | 5 Way | |
|---|---|---|
| | 1 Shot | 5 Shot |
| **miniImagenet** | | |
| MAML with GCP[#] | 46.92 ± 0.83 | 63.28 ± 0.66 |
| MAML with ATS[#] | 47.89 ± 0.77 | 64.07 ± 0.70 |
| MAML+UNIFORM (Offline)[#] | 46.67 ± 0.63 | 62.09 ± 0.55 |
| MAML+UNIFORM (Online)[#] | 46.70 ± 0.61 | 61.62 ± 0.54 |
| TA-MAML* (Ours) | 48.36 ± 0.23 | 62.48 ± 0.18 |
| **TA-MAML[#] (Ours)** | **53.80 ± 1.85** | **66.11 ± 0.11** |
| MetaSGD with GCP[#] | 47.77 ± 0.75 | 63.50 ± 0.71 |
| MetaSGD with ATS[#] | 48.59 ± 0.79 | 64.79 ± 0.74 |
| TA-MetaSGD* (Ours) | 49.28 ± 0.20 | 63.37 ± 0.16 |
| **TA-MetaSGD[#] (Ours)** | **52.60 ± 0.25** | **67.54 ± 0.12** |
| ANIL+UNIFORM (Offline)[#] | 46.93 ± 0.62 | 62.75 ± 0.60 |
| ANIL+UNIFORM (Online)[#] | 46.82 ± 0.63 | 62.63 ± 0.59 |
| TA-ANIL* (Ours) | 48.84 ± 0.62 | 60.80 ± 0.55 |
| **TA-ANIL[#] (Ours)** | **49.53 ± 0.41** | **63.73 ± 0.33** |
| **miniImagenet-noisy** | | |
| Uniform | 41.67 ± 0.80 | 55.80 ± 0.71 |
| SPL | 42.13 ± 0.79 | 56.19 ± 0.70 |
| Focal Loss | 41.91 ± 0.78 | 53.58 ± 0.75 |
| GCP | 41.86 ± 0.75 | 54.63 ± 0.72 |
| PAML | 41.49 ± 0.74 | 52.45 ± 0.69 |
| DAML | 41.26 ± 0.73 | 55.46 ± 0.70 |
| ATS | 44.21 ± 0.76 | 59.50 ± 0.71 |
| **TA-ANIL* (Ours)** | **45.17 ± 0.23** | **62.15 ± 1.01** |

proposed approach with GCP, ATS, and other sampling techniques on the miniImagenet-noisy dataset (Yao et al., 2021) and report the results in Table 3. We observe that task attention outperforms all scheduling algorithms on the miniImagenet-noisy dataset. As ATS is the most competitive baseline for the proposed method on the miniImagenet-noisy dataset, we compare the TA-ANIL and ATS on varying noise ratios for the miniImagenet dataset on 5 way 1 shot setting (Table 4). We observe that the proposed method outperforms ATS on all noise ratios except 0.8. Note that the algorithm used for all sampling approaches is ANIL.

## 5.4 Effectiveness of Task Attention in CDFSL setup

Classical meta-learning approaches assume meta-train and meta-test data belong to the same distribution such that the meta-trained model extends its knowledge to the meta-test set. This is, however, not always the case. The difference in the data acquisition techniques, or evolution

Table 4: Comparative analysis of ANIL integrated with ATS and proposed method on miniImagenet dataset with varying noise ratios for 5 way 1 shot setting. BNS is the best non-adaptive scheduler.

| Noise ratio | Test Accuracy (%) on miniImagenet-noisy | | | |
|---|---|---|---|---|
| | 0.2 | 0.4 | 0.6 | 0.8 |
| ANIL with Uniform | 43.46 ± 0.82 | 42.92 ± 0.78 | 41.67 ± 0.80 | 36.53 ± 0.73 |
| ANIL with BNS | 44.04 ± 0.81 | 43.36 ± 0.75 | 42.13 ± 0.79 | 38.21 ± 0.75 |
| ANIL with ATS | 45.55 ± 0.80 | 44.50 ± 0.86 | 44.21 ± 0.76 | **42.18 ± 0.73** |
| **TA-ANIL* (Ours)** | **47.98 ± 0.26** | **46.69 ± 0.22** | **45.17 ± 0.23** | 40.35 ± 1.14 |

of data with time, may cause a discrepancy between the meta-train and meta-test distributions. This realistic setting is popularly termed as cross-domain few-shot learning (CDFSL) (Guo et al., 2020). We conducted experiments to show the merit of the proposed approach in CDFSL setup. Specifically, we train a model using a TA meta-training regimen on the miniImagenet dataset and meta-test it on CUB-200, FGVC-Aircraft, Describable Textures, and Omniglot datasets from Metadataset (Triantafillou et al., 2019). The results reported for 5 way 1 and 5 shot settings in Table 5 indicate that the proposed approach outperforms the state-of-the-art task scheduling approach (Uniform Sampling - wherever applicable) or non-task-attended counterparts (for Omniglot) on CDFSL setup by a large margin.

As some classes of Imagenet overlap with Metadataset, we also conduct experiments on the diverse VTAB dataset (Zhai et al., 2019), which does not share classes with the Imagenet (consequently miniImagenet) dataset. We note that some VTAB sub-datasets like Sun397 are quite memory intensive and others like Patch Camelyon, Retinopathy, etc., have fewer classes. In the interest of time and resources, we meta-train a conv4 model on the miniImagenet dataset and evaluate it on a few of feasible sub-datasets covering all three domains - Natural, Specialized, and Structured. Specif-

Table 5: Comparative analysis of proposed approach (TA-MAML) and uniform sampling (Arnold et al., 2021) (or non-task attended counterpart (MAML)) in a CDFSL setting after training on miniImagenet dataset and tested on Metadataset and VTAB datasets for 5 way 1 and 5 shot settings.

| Model | 5 Way | | 5 Way | |
|---|---|---|---|---|
| | 1 Shot | 5 Shot | 1 Shot | 5 Shot |
| | **Metadataset** | | | |
| | **CUB-200** | | **FGVC-Aircraft** | |
| MAML+ UNIFORM (Online)# | 35.84 ± 0.54 | 46.67 ± 0.55 | 26.62 ± 0.39 | 34.41 ± 0.44 |
| **TA-MAML# (Ours)** | **42.87 ± 1.18** | **57.49 ± 0.99** | **29.42 ± 0.78** | **36.34 ± 0.86** |
| | **Describable Textures** | | | |
| MAML+ UNIFORM (Online)# | 31.84 ± 0.49 | 40.81 ± 0.44 | | |
| **TA-MAML# (Ours)** | **31.98 ± 0.98** | **44.39 ± 0.79** | | |
| | **Omniglot** | | | |
| MAML# | 72.40 ± 1.43 | 86.81 ± 0.99 | | |
| **TA-MAML#(Ours)** | **78.73 ± 1.08** | **88.92 ± 0.76** | | |
| | **VTAB Dataset** | | | |
| | **FC100** | | **Flowers102** | |
| MAML# | 35.49 ± 1.95 | 44.42 ± 0.83 | 51.93 ± 1.59 | 75.22 ± 0.48 |
| **TA-MAML# (Ours)** | **38.87 ± 1.90** | **46.57 ± 0.85** | **61.86 ± 1.72** | **77.49 ± 0.16** |
| | **SVHN** | | | |
| MAML# | 20.93 ± 1.01 | 22.42 ± 0.88 | | |
| **TA-MAML# (Ours)** | **21.73 ± 1.09** | **24.20 ± 0.78** | | |
| | **EuroSAT** | | **Resisc45** | |
| MAML# | 45.80 ± 1.49 | 62.0 ± 0.71 | 33.60 ± 1.49 | 42.07 ± 0.37 |
| **TA-MAML# (Ours)** | **51.67 ± 1.62** | **66.69 ± 0.70** | **35.20 ± 1.21** | **46.27 ± 0.39** |
| | **DSprites_location** | | **DSprites_orientation** | |
| MAML# | 36.67 ± 1.55 | 48.91 ± 0.84 | 20.86 ± 1.81 | 22.89 ± 0.95 |
| **TA-MAML# (Ours)** | **39.93 ± 1.33** | **56.48 ± 0.95** | **24.27 ± 1.18** | **22.92 ± 0.93** |

ically, we investigate the merit of the proposed approach on Natural sub-datasets like DTD, CIFAR FC 100, Flowers102, and SVHN, specialized sub-datasets like EuroSAT and Resisc45, and structured sub-datasets like dSprites_location and dSprites_orientation. We have kept Describable Textures as a part of Metadataset and Flowers102 as a component of VTAB dataset according to (Dumoulin et al., 2021). We convert the selected VTAB sub-datasets to a few-shot setup (5-way 1 and 5 shot tasks) and evaluate task-attended MAML (TA-MAML) and its vanilla version (MAML) on 300 tasks. Our experiments (Table 5) demonstrate that task attention allows MAML to better generalize to unseen, diverse out-of-distribution VTAB meta-test sets.

## 5.5 Ablation Studies

To examine the significance of each input given to the task attention model, we conduct an ablation study on 5 way 1 and 5 shot TA-MAML on miniImagenet dataset and report the results in Table 6. We observe that all the components of meta-information contribute to the learning of a more generalizable meta-model. To further support this observation, we investigate the relationship between the meta-information and weights assigned by the task attention module by analyzing the mean Pearson correlation of

Table 6: Effect of ablating components of meta-information in TA-MAML$^*$ for 5 way 1 and 5 shot settings on miniImagenet dataset.

| Ablation on inputs | | | | Test Accuracy | |
|---|---|---|---|---|---|
| Grad norm | Loss | Loss-ratio | Accuracy | 5 way 1 shot | 5 way 5 shot |
| × | × | × | × | 46.10±0.19 | 60.16±0.17 |
| ✓ | ✓ | ✓ | × | 47.30±0.16 | 60.48±0.16 |
| ✓ | ✓ | × | ✓ | 47.62±0.17 | 62.17±0.17 |
| ✓ | × | ✓ | ✓ | 48.10±0.18 | 60.90±0.20 |
| × | ✓ | ✓ | ✓ | 47.30±0.18 | 61.52±0.16 |
| ✓ | ✓ | ✓ | ✓ | **48.36±0.23** | **62.48±0.18** |

each of the components (four tuple) of the meta-information with the attention vector across the training iterations. This is depicted in Figure 4 for TA-MAML on 5 way 1 and 5 shot settings for miniImagenet dataset. We observe that the loss ratio and loss are positively correlated with the attention vector, while accuracy and gradient norm are negatively correlated.

In 5 way 5 shot setting, we observe that the correlation pattern is comparable to 5 way 1 shot setting, but the mean correlation value of grad norm across iterations is less than that of the 5 way 1 shot setting. This could be because the 5 way 5 shot setting is richer in data than the 5 way 1 shot setting, which allows better learning and therefore has low average values of grad norm (Section 4.1.1). The critical observation, however, is that the meta-information components have a weak correlation with the attention weights, indicating that the TA module does not trivially follow any single component of meta-information. We also analyze the ranks of the tasks for maximum and minimum values of : loss, loss ratio, accuracy, and grad norm in a batch, as per the weights across training iterations, and describe results in the supplementary material. The rank analysis also reinforces the same observation. We ascertain the decreasing trend of mean weighted loss across iterations in the supplementary material.

## 5.6 Analysis of Attention Network

To gain further insights into the operation of the attention module, we also examine the trend of the attention-vector (Figure 5) while meta-training TA-MAML for 5 way 1 and 5 shot settings on the miniImagenet dataset. We plot the maximum and the minimum attention score assigned to the tasks of a batch across iterations together with a few weighted task batches in 5 way 1 shot setting for illustration. We note that the weighted task batches are only intended

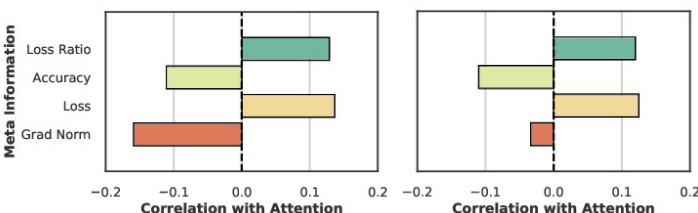

Figure 4: Mean Pearson correlation of TA-MAML$^*$ on 5 way 1 shot (left) and 5 shot (right) setting on miniImagenet.

to demonstrate the change in the tasks' attention scores across iterations. The next experiment presents a more rigorous analysis studying the relationship among classes in a task and attention scores assigned.

We note that the mean attention score is always 0.25 as we follow a meta-batch size of 4. We observe that the TA module's output follows an interesting trend. Initially, the TA module assigns almost uniform weights to all the tasks of a batch; however, as the iterations increase, it assigns unequal scores to the tasks in a batch, preferring some over the other. This suggests that during the initial phases of the meta-model's training, all tasks have equal contribution towards learning a *generic structure* of the meta-knowledge. As the meta-model's learning proceeds, learning the further *fine-grained meta-knowledge structure* requires prioritizing some tasks in a batch over the others, which are potentially better aligned with learning the optimal meta-knowledge. We study the computational burden imposed by TA regimen in the appendix.

We further decipher the functioning of the black box attention network by analyzing the qualitative relation among weights and the classes of task batches (Figure 6). In Figure 6 left column (col-1) corresponds to the cases where the assignment of attention scores to the tasks is human interpretable. In contrast, the right column (col-2) refers to the uninterpretable attention scores. From the human perspective, tasks containing images from similar classes are hard to distinguish and are assigned higher attention scores indicated by red bounding boxes (Figure 6 col-1). Specifically, (col-1, row-1) task 2 is regarded as most important,

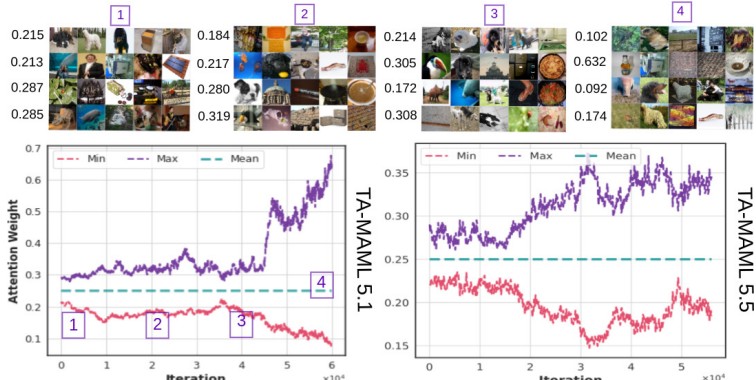

Figure 5: Trend of an attention vector in 5 way 1 shot (left) and 5 shot (right) settings on miniImagenet dataset for TA-MAML*.

possibly because it includes three breeds of dogs followed by task 4, which comprises two species of fish. However, the aforementioned is not a hard constraint, as there are some task batches (Figure 6 col-2) in which the distribution of weights cannot be explained qualitatively.

## 6 Conclusion

In this work we have shown that the batch wise episodic training regimen adopted by ML strategies can benefit from leveraging knowledge about the importance of tasks within a batch. Unlike prior approaches that assume uniform importance for each task in a batch, we propose task attention as a way to learn the relevance of each task according to its alignment with the optimal meta-knowledge. We have validated the effectiveness of task attention by augmenting it to popular initialization and optimization based ML strategies. We have demonstrated through experiments on miniImagenet, FC100 and tieredImagenet datasets that augmenting task attention helps attain better generalization to unseen tasks from the same distribution while requiring fewer iterations to converge. We also show that the task attention is meritorious over existing task scheduling algorithms, even on noisy and CDFSL setups. We also conduct an exhaustive empirical analysis on the distribution of attention weights to study the nature of the meta-knowledge and task attention module. We leave the theoretical motivation of the meta-information components and the proof of convergence of the proposed curriculum as part of our future work. We believe that this end-to-end attention-based meta training paves the way towards efficient and automated meta-training.

## 7 Broader Impact

We acknowledge that transfer and metric approaches like (Kolesnikov et al., 2020; Triantafillou et al., 2019; Bronskill et al., 2021; Dvornik et al., 2020) use more advanced backbones and our approach is limited to a basic architecture (Conv4) and gradient-based methods. We clarify that though our approach is extendable to any episodic curriculum (including metric approaches with minor design changes), we choose gradient-based approaches like MAML and ANIL approaches as they are domain-agnostic in contrast to metric learning. However, we leave the investigation of attention mechanisms for metric approaches and domains, such as reinforcement learning or regression problems for gradient approaches for future work. Unfortunately, due

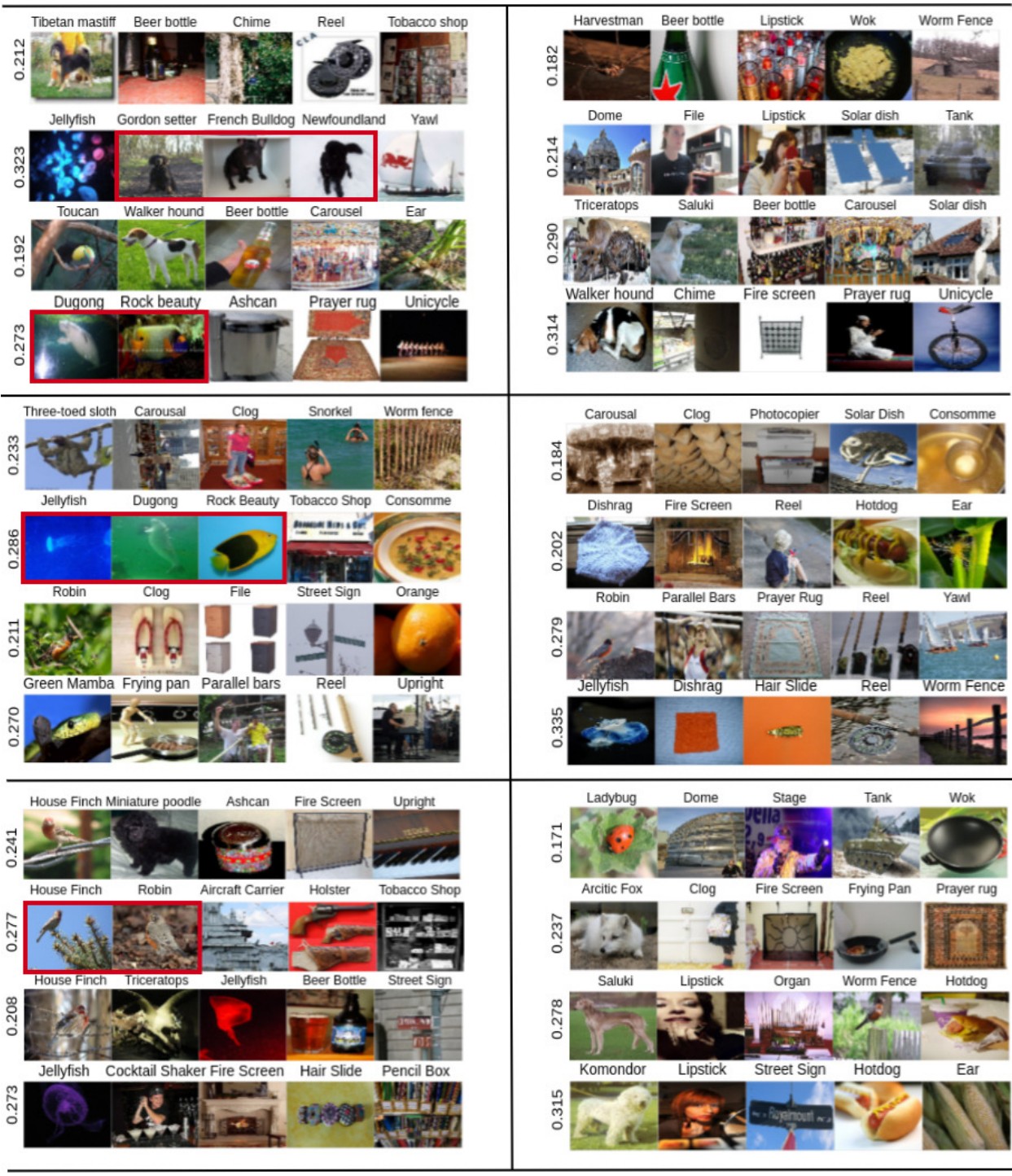

Figure 6: Explanations of TA module in TA-MAML* on miniImagenet. **Left Col)** Higher weights accredited to tasks with comparable classes marked by red bounding boxes. **Right Col)** Association of weights and task data is qualitatively uninterpretable. Rows correspond to the batches.

to computational and storage restrictions, we are unable to experiment with deeper backbones and large image sizes for gradient-based methods. We, therefore, limit the scope of our study only to algorithms,

datasets, and conditions and leave the scalability aspect to the future. We, however, point out the existing literature (Chen et al., 2018) that compares vanilla transfer learning (with no Imaganet pretraining or data augmentation) for conv4 backbone with episodic training (MAML) under fair conditions. Chen et al. have demonstrated that MAML performs better than vanilla transfer learning under fair conditions for conv4 architecture. However, transfer learning scales much better with the architectures than MAML (or other episodic methods) (Chen et al., 2018). Nevertheless, transfer learning (TL) is a good solution for few-shot learning (especially with Imagenet pretraining and larger backbones), and translating attention to TL for a few-shot setup is a promising direction for further research. An attention module, in this case, could be used to reweigh the examples instead of tasks, and it could be trained using a smaller validation data pool. Also, sampling a validation pool from a combination of distributions (transduction) is worth exploring. We leave these extensions for future work. We, acknowledge, that similar to (Yao et al., 2021; Wu et al., 2022; Raghu et al., 2020), our study is limited to understanding the fundamentals of episodic training rather than developing an algorithm that surpasses the state-of-the-art approach for few shot learning.

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

## 8 Appendix

### 8.1 Preliminary

#### 8.1.1 Meta-knowledge as an Optimal Initialization

When meta-knowledge is a generic initialization on the model parameters learned through the experience over various tasks, it is enforced to be close to each individual training tasks' optimal parameters. A model initialized with such an optimal prior quickly adapts to unseen tasks from the same distribution during meta-testing. **MAML** (Finn et al., 2017) employs a nested iterative process to learn the task-agnostic optimal prior $\theta$. In the inner iterations representing the task adaptation steps, $\theta$ is separately fine-tuned for each meta-training task $\mathcal{T}_i$ of a batch using $D_i$ to obtain $\phi_i$ through gradient descent on the train loss $L$ using learning rate $\alpha$. Specifically, $\phi_i$ is initialized as $\theta$ and updated using $\phi_i \leftarrow \phi_i - \alpha \nabla_{\phi_i} L(\phi_i)$, $T$ times resulting in the adapted model $\phi_i^T$. In the outer loop, meta-knowledge is gathered by optimizing $\theta$ over loss $L^*$ computed with the task adapted model parameters $\phi_i^T$ on query dataset $D_i^*$. Specifically, during meta-optimization $\theta \leftarrow \theta - \beta \nabla_\theta \sum_{i=1}^B L^*(\phi_i^T)$ using a task batch of size $B$ and learning rate $\beta$. **MetaSGD** (Li et al., 2017) improves upon MAML by learning parameter-specific learning rates $\boldsymbol{\alpha}$ in addition to the optimal initialization in a similar nested iterative procedure. Meta-knowledge is gathered by optimizing $\theta$ and $\boldsymbol{\alpha}$ in the outer loop using the loss $L^*$ computed on query set $D_i^*$. Specifically, during meta-optimization $(\theta, \boldsymbol{\alpha}) \leftarrow (\theta, \boldsymbol{\alpha}) - \beta \nabla_{(\theta, \boldsymbol{\alpha})} \sum_{i=1}^B L^*(\phi_i^T)$. Learning dynamic learning rates for each parameter of a model makes MetaSGD faster and more generalizable than MAML. A single adaptation step is sufficient to adjust the model towards a new task. The performance of MAML is attributed to the reuse of the features across tasks rather than the rapid learning of new tasks (Raghu et al., 2020). Exploiting this characteristic, **ANIL** freezes the feature backbone layers $(1, \ldots, l-1)$ and only adapts classifier layer $(l)$ in the inner loop $T$ times. Specifically during adaptation $\phi_i^l \leftarrow \phi_i^l - \alpha \nabla_{\phi_i^l} L(\phi_i^l)$. During meta-optimization $\theta^{1,\ldots,l} \leftarrow \theta^{1,\ldots,l} - \beta \nabla_{\theta^{1,\ldots,l}} \sum_{i=1}^B L^*(\phi_i^{lT})$ i.e., all layers are learned in the outer loop. Freezing the feature backbone during adaptation reduces the overhead of computing gradient through the gradient (differentiating through the inner loop), and thereby heavier backbones could be used for the feature extraction. **TAML** (Jamal & Qi, 2019) suggests that the optimal prior learned by MAML may still be biased towards some tasks. They propose to reduce this bias and enforce equity among the tasks by explicitly minimizing the inequality among the performances of tasks in a batch. The inequality defined using statistical measures such as Theil Index, Atkinson Index, Generalized Entropy Index, and Gini Coefficient among the performances of tasks in a batch is used as a regularizer while gathering the meta-knowledge. For the baseline comparison, in our experiments, we use the Theil index for TAML owing to its average best results. Specifically during meta-optimization $\theta \leftarrow \theta - \beta \nabla_\theta \left[ \sum_{i=1}^B L^*(\phi_i^T) + \lambda \left\{ \frac{L^*(\phi_i^0)}{\bar{L}^*(\phi_i^0)} \ln \frac{L^*(\phi_i^0)}{\bar{L}^*(\phi_i^0)} \right\} \right]$ (for TAML-Theil Index) where $B$ is the number of tasks in a batch, $L^*(\phi_i^0)$ is the loss incurred by initial model $\phi_i^0$ on the query set $D_i^*$ of task $\mathcal{T}_i$ and $\bar{L}^*(\phi_i^0)$ is the average query loss of initial model on a batch of tasks. As TAML enforces equity of the optimal prior towards meta-train tasks, it counters the adaptation, which leads to slow and unstable training largely dependent on $\lambda$.

#### 8.1.2 Meta-knowledge as a Parametric Optimizer

A regulated gradient-based optimizer gathers the task-specific and task-agnostic meta-knowledge to traverse the loss surfaces of tasks in the meta-train set during meta-training. A base model guided by such a learned parametric optimizer quickly finds the way to minima even for unseen tasks sampled from the same distribution during meta-testing. **MetaLSTM** (Ravi & Larochelle, 2017) is a recurrent parametric optimizer $\theta$ that mimics the gradient-based optimization of a base model $\phi$. This recurrent optimizer is an LSTM (Hochreiter & Schmidhuber, 1997) and is inherently capable of performing two-level learning due to its architecture. During adaptation of $\phi_i$ on $D_i$, $\theta$ takes meta information of $\phi_i$ characterized by its current loss $L$ and gradients $\nabla_{\phi_i}(L)$ as input and outputs the next set of parameters for $\phi_i$. This adaptation procedure is repeated $T$ times resulting in the adapted base-model $\phi_i^T$. Internally, the cell state of $\theta$ corresponds to $\phi_i$, and the cell state update for $\theta$ resembles a learned and controlled gradient update. The emphasis on previous parameters and the current update is regulated by the learned forget and input gates respectively. While

adapting $\phi_i$ to $D_i$, information about the trajectory on the loss surface across the adaptation steps is captured in the hidden states of $\theta$, representing the task-specific knowledge. During meta-optimization, $\theta$ is updated based on the loss of the adapted model $L^*(\phi_i^T)$ computed on the query set $D_i^*$ to garner the meta-knowledge across tasks. Specifically, during meta-optimization, $\theta \leftarrow \theta - \beta \nabla_\theta L^*(\phi_i^T)$. MetaLSTM updates parametric optimizer $\theta$ after adapting the base model $\phi$ to each task. This causes $\theta$ to follow optima's of all adapted base models leading to its elongated and fluctuating optimization trajectory, which is biased towards the last task. **MetaLSTM++** (Aimen et al., 2021) circumvents these issues as $\theta$ is updated by an aggregate query loss of the adapted models on a batch of tasks. Batch updates smoothen the optimization trajectory of $\theta$ and eliminate its bias towards the last task. Specifically, during meta-optimization $\theta \leftarrow \theta - \beta \nabla_\theta \sum_{i=1}^{B} L^*(\phi_i^T)$.

## 8.2 Detailed Explanation of the Proposed approach

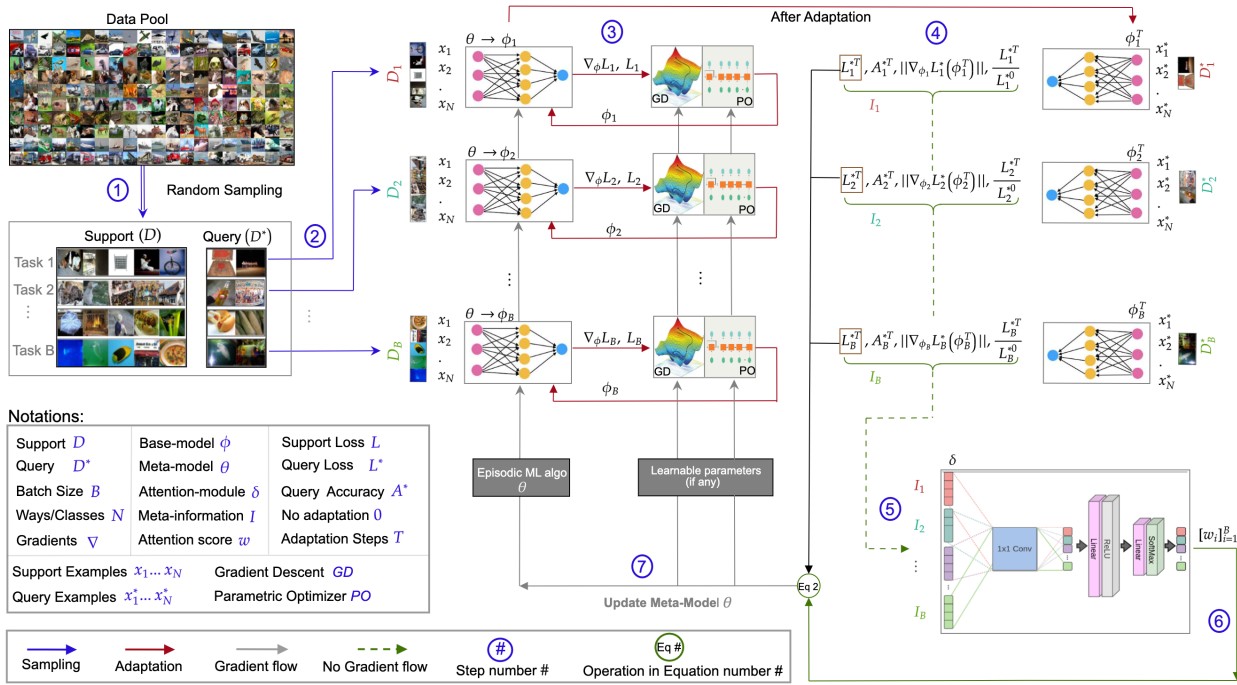

Figure 7: [Best viewed in color] Workflow of proposed training curriculum.

We explain the proposed approach through Figure 1, Figure 7, Algorithm 1, and equations. We first sample a batch of tasks ($B$) from a random pool of data (Figure 7 - Label ①). For each task, the base-model $\phi_i$ is adapted using the support data $D_i$ for $T$ time-steps (line 7 and lines 20-32 in Algorithm 1, Figure 7 - Label ③). Specifically, the adaptation is done using gradient descent on the train loss $L$ for initialization approaches (lines 22-26 in Algorithm 1, Figure 7 - GD), or the current loss and gradients are inputted to the meta-model $\theta$ for optimization approaches, which then outputs the updated base-model parameters (lines 27-31 in Algorithm 1, Figure 7 - PO). The meta-information ($\mathcal{I}$) corresponding to each task in the batch is then calculated (Figure 7 - Label ④), which includes the loss, accuracy, loss-ratio, and gradient norm of adapted models on the query data. This is given as input to the task attention module (Figure 1 - Label ②, Figure 7 - Label ⑤), which outputs the attention vector (line 10 in Algorithm 1, Figure 7- Label ⑥). The attention vector and test losses are used to update the meta-model parameters $\theta$ according to equation 2 (line 11 in Algorithm 1, Figure 1 - Label ④, Figure 7 - Label ⑦). A new batch of tasks is then sampled and the base-models are adapted using the updated meta-model (Lines 12-16 in Algorithm 1, Figure 1 - Label ⑤). The mean test loss over the adapted base-models is calculated and used to update the parameters of the task attention module $\delta$ according to equation 3.

### 8.3 Experiments

### 8.3.1 Datasets Details

**miniImagenet** dataset (Vinyals et al., 2016) comprises 600 color images of size 84 × 84 from each of 100 classes sampled from the Imagenet dataset. The 100 classes are split into 64, 16 and 20 classes for meta-training, meta-validation and meta-testing respectively. **miniImagenet-noisy** (Yao et al., 2021) is constructed from the miniImagenet dataset with the additional constraint that tasks have noisy support labels and clean query labels. The noise in support labels is introduced by symmetry flipping, and the default noise ratio is 0.6. **Fewshot Cifar 100 (FC100)** dataset (Oreshkin et al., 2018) has been created from Cifar 100 object classification dataset. It contains 600 color images of size 32 × 32 corresponding to each of 100 classes grouped into 20 super-classes. Among 100 classes, 60 classes belonging to 12 super-classes correspond to the meta-train set, 20 classes from 4 super-classes to the meta-validation set, and the rest to the meta-test set. **tieredImagenet** (Ren et al., 2018a) is a more challenging benchmark for few-shot image classification. It contains 779,165 color images sampled from 608 classes of Imagenet and are grouped into 34 super-classes. These super-classes are divided into 20, 6, and 8 disjoint sets for meta-training, meta-validation, and meta-testing. **Metadataset** (Triantafillou et al., 2019) comprises of 10 freely available diverse datasets - Aircraft, CUB-200-2011, Describable Textures, Fungi, ILSVRC-2012, MSCOCO, Omniglot, Quick Draw, Traffic Signs, and VGG Flower. We utilized CUB-200, FGVC-Aircraft, Describable Textures, and Omniglot datasets from Metadataset. **VTAB dataset** (Zhai et al., 2019) is a more diverse dataset than Metadataset that was proposed to avoid overlapping classes of sub-datasets with the Imagenet dataset. VTAB comprises of 19 datasets divided into three domains - Natural, Specialized, and Structured, depending on the type of images. The natural group contains Caltech101, CIFAR100, DTD, Flowers102, Pets, Sun397, and SVHN sub-datasets, while the specialized group consists of remote sensing datasets like EuroSAT and Resisic 45 and medical datasets like Retinopathy and Patch Camelyon. Structured contains object counting or 3D depth prediction datasets like Clevr/count, Clevr/distance, dSprites/location, dSprites/orientation, Small-NORB/azimuth, SmallNORB/elevation, DMLab, and KITTI/distance. We considered Natural sub-datasets like DTD, CIFAR FC 100, Flowers102, and SVHN, specialized sub-datasets like EuroSAT and Resisc45, and structured sub-datasets like dSprites_location and dSprites_orientation for cross-domain experimentation. According to (Dumoulin et al., 2021), we have kept Describable Textures as a part of Metadataset and Flowers102 as a component of the VTAB dataset.

### 8.3.2 Ablation Studies

We analyze the ranks of the tasks for maximum and minimum values of : loss, loss ratio, accuracy, and grad norm in a batch wrt attention weights throughout meta-training of TA-MAML on a 5 way 1 and 5 shot settings on miniImagenet dataset (Figures 8 and 9). Specifically, the highest weighted task is given rank one, and the least weighted task in a batch is given the last rank. We observe that the TA module does not assign maximum weight to the tasks with maximum or minimum values of : test loss, loss ratio, grad norm or accuracy throughout meta-training. Thus, the TA module does not trivially learn to assign weights to the tasks based on some component of meta-information but learns useful latent information from all the components to assign importance for the tasks in a batch.

### 8.3.3 Relation of Weights with Meta-Information

In Figure 10, we illustrate the trend of mean weighted loss across iterations for TA-MAML on 5 way 1 and 5 shot settings on miniImagenet dataset. The trend indicates that the average weighted loss decreases over the meta-training iterations. The shaded region represents a 95% confidence interval over 100 tasks.

### 8.3.4 Computational Overhead

The training time for all scheduling/sampling approaches is expected to be higher than their non-scheduling/sampling counterparts. We observe a three-fold increase in the training time from the vanilla setting for a model trained with our strategy and a two-fold increase in the training time if a non-neural scheduling approach (Liu et al., 2021a) is employed. However, our approach significantly outperforms vanilla

**5 way 1 shot setting**

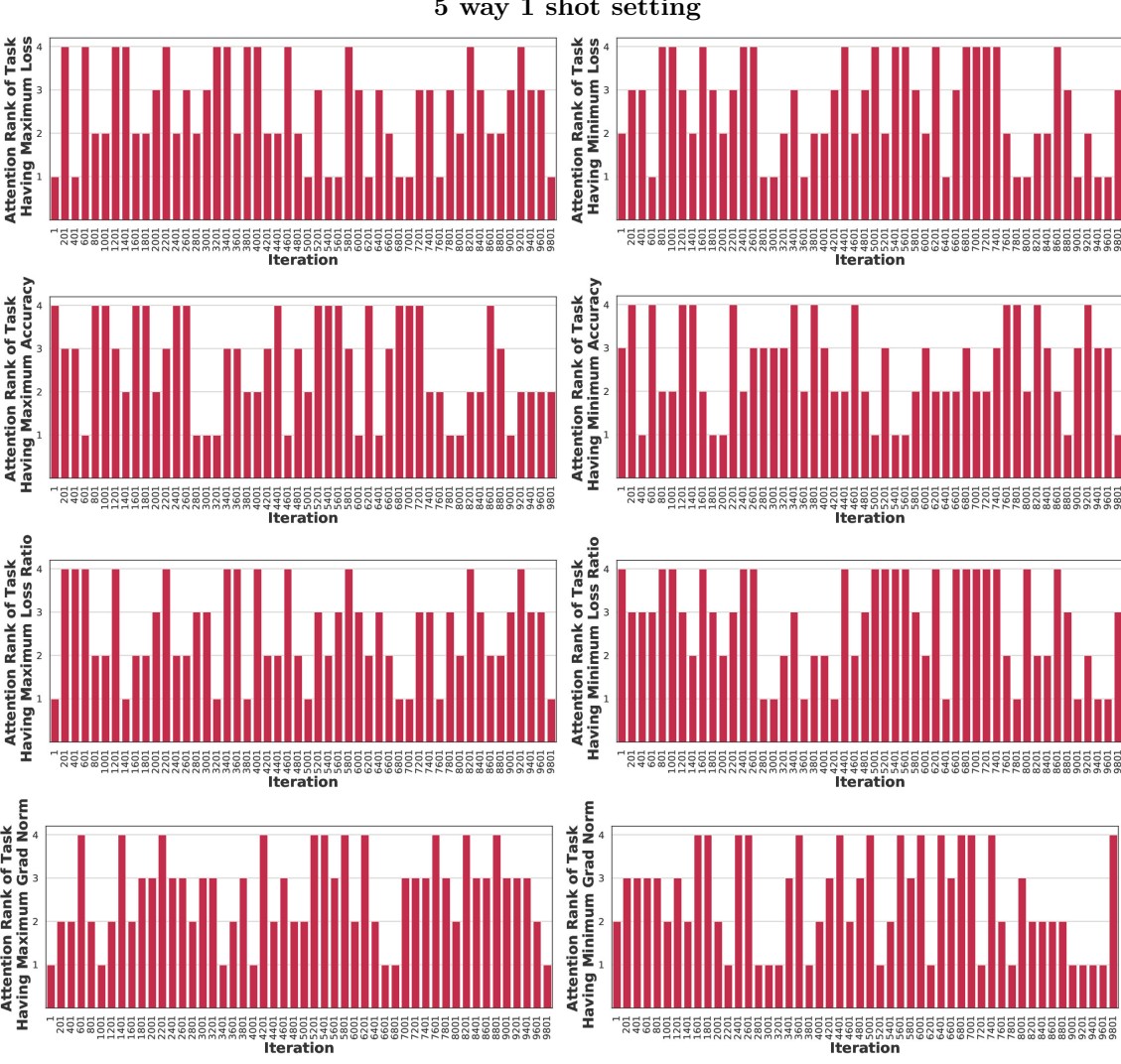

Figure 8: Rank Analysis of tasks for maximum and minimum values of : loss, loss-ratio, accuracy and grad norm throughout the training of TA-MAML* for 5 way 1 shot setting on miniImagenet dataset.

Table 7: Comparison of few-shot classification performance of MAML and ANIL reported in the original papers (denoted by #) and the re-implementation by others on miniImagenet dataset for 5 way 1 and 5 shot settings. The highest and lowest accuracies for an approach are represented in blue and red, respectively.

| | Test Accuracy (%) | |
| --- | --- | --- |
| | 5 Way | |
| Model | 1 Shot | 5 Shot |
| | miniImagenet | |
| MAML#(Finn et al., 2017) | 48.07 ± 1.75 | 63.15 ± 0.91 - |
| MAML (Antoniou et al., 2019) | 48.25 ± 0.62 | 64.39 ± 0.31 |
| MAML (Raghu et al., 2020) | 46.9 ± 0.2 | 63.1 ± 0.4- |
| MAML (Chen et al., 2018) | 46.47 ± 0.82 | 62.71 ± 0.71 |
| MAML(Oh et al., 2020) | 47.44 ± 0.23 | 61.75 ± 0.42 |
| MAML (Agarwal et al., 2021) | 47.13 ± 8.78 | 57.69 ± 7.92 |
| MAML (Arnold et al., 2021) | 46.88 ± 0.60 | 55.16 ± 0.55 |
| ANIL#(Raghu et al., 2020) | 46.7 ± 0.4 | 61.5 ± 0.5 |
| ANIL(Oh et al., 2020) | 47.82 ± 0.20 | 63.04 ± 0.42 |
| ANIL(Arnold et al., 2021) | 46.59±0.60 | 63.47±0.55 |

**5 way 5 shot setting**

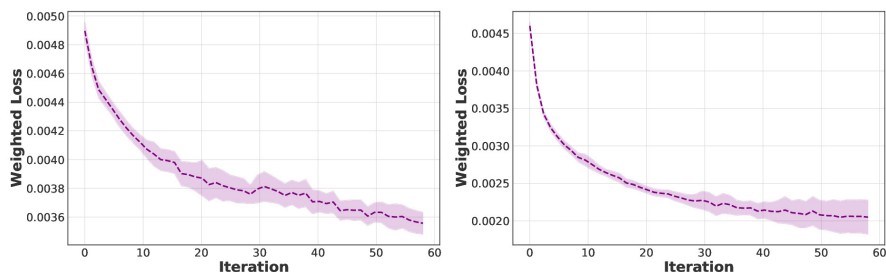

Figure 9: Rank Analysis of tasks for maximum and minimum values of : loss, loss-ratio, accuracy and grad norm throughout the training of TA-MAML* for 5 way 5 shot setting on miniImagenet dataset.

Figure 10: Trend analysis of weighted loss across meta-training iterations for TA-MAML* on 5 way 1 shot (left) and 5 shot (right) settings on miniImagenet dataset. Iterations are in thousands.

ML approaches and all state-of-the-art scheduling strategies on various datasets, training setups, and learning paradigms (Tables 2, 3, 4 and 5). As training is typically performed offline, the increased computational overhead is expected to be permissible. Further, ours, as well as other approaches, perform vanilla finetuning during meta-testing (i.e., task attention, neural scheduling or conflict resolving mechanism is not employed during meta-testing), resulting in comparable test time (15-20 seconds on 300 tasks for MAML 5-way 1- and 5-shot setups). We also note that we do not pre-train the attention network, unlike state-of-the-art schedulers like ATS.

849 **8.3.5 Hyperparameter Details**

| Setting | Model | base lr | meta lr | attention lr | lambda |
|---------|-------|---------|---------|--------------|--------|
| | | **miniImagenet** | | | |
| 5.1 | MAML | 0.5000 | 0.0030 | - | - |
| | TAML | 0.5000 | 0.0030 | - | 0.0748 |
| | TA-MAML* | 0.0763 | 0.0005 | 0.0004 | - |
| | MetaSGD | 0.5000 | 0.0030 | - | - |
| | TA-MetaSGD* | 0.0529 | 0.0011 | 0.0004 | - |
| | MetaLSTM | - | 0.005 | - | - |
| | MetaLSTM++ | - | 0.0012 | - | - |
| | TA-MetaLSTM++* | - | 0.0012 | 0.0031 | - |
| | ANIL | 0.3000 | 0.0006 | - | - |
| | TA-ANIL* | 0.0763 | 0.0005 | 0.0004 | - |
| 5.5 | MAML | 0.5000 | 0.0030 | - | - |
| | TAML | 0.5000 | 0.0030 | - | 0.7916 |
| | TA-MAML* | 0.0763 | 0.0005 | 0.0004 | - |
| | MetaSGD | 0.5000 | 0.0030 | - | - |
| | TA-MetaSGD* | 0.0529 | 0.0011 | 0.0004 | - |
| | MetaLSTM | - | 0.005 | - | - |
| | MetaLSTM++ | - | 0.0012 | - | - |
| | TA-MetaLSTM++* | - | 0.0004 | 0.0001 | - |
| | ANIL | 0.3000 | 0.0006 | - | - |
| | TA-ANIL* | 0.0763 | 0.0005 | 0.0004 | - |
| 10.1 | MAML | 0.5000 | 0.0030 | - | - |
| | TAML | 0.5000 | 0.0030 | - | 0.2631 |
| | TA-MAML* | 0.2551 | 0.0015 | 0.0001 | - |
| | MetaSGD | 0.5000 | 0.0030 | - | - |
| | TA-MetaSGD* | 0.0627 | 0.0008 | 0.0013 | - |
| | MetaLSTM | - | 0.005 | - | - |
| | MetaLSTM++ | - | 0.0015 | - | - |
| | TA-MetaLSTM++* | - | 0.0009 | 0.0015 | - |
| | ANIL | 0.5000 | 0.0030 | - | - |
| | TA-ANIL* | 0.2551 | 0.0015 | 0.0001 | - |
| 10.5 | MAML | 0.5000 | 0.0030 | - | - |
| | TAML | 0.5000 | 0.0030 | - | 0.0741 |
| | TA-MAML* | 0.2551 | 0.0015 | 0.0001 | - |
| | MetaSGD | 0.5000 | 0.0030 | - | - |
| | TA-MetaSGD* | 0.0627 | 0.0008 | 0.0013 | - |
| | MetaLSTM | - | 0.005 | - | - |
| | MetaLSTM++ | - | 0.0036 | - | - |
| | TA-MetaLSTM++* | - | 0.0024 | 0.0002 | - |
| | ANIL | 0.5000 | 0.0030 | - | - |
| | TA-ANIL* | 0.2551 | 0.0015 | 0.0001 | - |

| Setting | Model | base lr | meta lr | attention lr | lambda |
|---|---|---|---|---|---|
| | | | **FC100** | | |
| 5.1 | MAML | 0.5000 | 0.0030 | - | - |
| | TAML | 0.5000 | 0.0030 | - | 0.0164 |
| | TA-MAML* | 0.2826 | 0.0003 | 0.0024 | - |
| | MetaSGD | 0.5000 | 0.0030 | - | - |
| | TA-MetaSGD* | 0.0349 | 0.0008 | 0.0001 | - |
| | MetaLSTM | - | 0.005 | - | - |
| | MetaLSTM++ | - | 0.0010 | - | - |
| | TA-MetaLSTM++* | - | 0.0002 | 0.0074 | - |
| | ANIL | 0.5000 | 0.0030 | - | - |
| | TA-ANIL* | 0.2826 | 0.0003 | 0.0024 | - |
| 5.5 | MAML | 0.5000 | 0.0030 | - | - |
| | TAML | 0.5000 | 0.0030 | - | 0.0153 |
| | TA-MAML* | 0.2826 | 0.0003 | 0.0024 | - |
| | MetaSGD | 0.5000 | 0.0030 | - | - |
| | TA-MetaSGD* | 0.0349 | 0.0008 | 0.0001 | - |
| | MetaLSTM | - | 0.005 | - | - |
| | MetaLSTM++ | - | 0.0002 | - | - |
| | TA-MetaLSTM++* | - | 0.0007 | 0.0003 | - |
| | ANIL | 0.5000 | 0.0030 | - | - |
| | TA-ANIL* | 0.2826 | 0.0003 | 0.0024 | - |
| 10.1 | MAML | 0.5000 | 0.0030 | - | - |
| | TAML | 0.5000 | 0.0030 | - | 0.0794 |
| | TA-MAML* | 0.2353 | 0.0002 | 0.0001 | - |
| | MetaSGD | 0.5000 | 0.0030 | - | - |
| | TA-MetaSGD* | 0.2583 | 0.0029 | 0.0007 | - |
| | MetaLSTM | - | 0.005 | - | - |
| | MetaLSTM++ | - | 0.0021 | - | - |
| | TA-MetaLSTM++* | - | 0.0005 | 0.0014 | - |
| | ANIL | 0.5000 | 0.0030 | - | - |
| | TA-ANIL* | 0.2826 | 0.0003 | 0.0024 | - |
| 10.5 | MAML | 0.5000 | 0.0030 | - | - |
| | TAML | 0.5000 | 0.0030 | - | 0.0193 |
| | TA-MAML* | 0.2353 | 0.0002 | 0.0001 | - |
| | MetaSGD | 0.5000 | 0.0030 | - | - |
| | TA-MetaSGD* | 0.2583 | 0.0029 | 0.0007 | - |
| | MetaLSTM | - | 0.005 | - | - |
| | MetaLSTM++ | - | 0.0004 | - | - |
| | TA-MetaLSTM++* | - | 0.0004 | 0.0090 | - |
| | ANIL | 0.5000 | 0.0030 | - | - |
| | TA-ANIL* | 0.2826 | 0.0003 | 0.0024 | - |

| Setting | Model | base lr | meta lr | attention lr | lambda |
|---------|-------|---------|---------|--------------|--------|
| | | | **tieredImagenet** | | |
| 5.1 | MAML | 0.5000 | 0.0030 | - | - |
| | TAML | 0.5000 | 0.0030 | - | 0.3978 |
| | TA-MAML* | 0.0261 | 0.0005 | 0.0015 | - |
| | MetaSGD | 0.5000 | 0.0030 | - | - |
| | TA-MetaSGD* | 0.0944 | 0.0003 | 0.0002 | - |
| | MetaLSTM | - | 0.005 | - | - |
| | MetaLSTM++ | - | 0.0002 | - | - |
| | TA-MetaLSTM++* | - | 0.0010 | 0.0006 | - |
| | ANIL | 0.5000 | 0.0030 | - | - |
| | TA-ANIL* | 0.0261 | 0.0005 | 0.0015 | - |
| 5.5 | MAML | 0.5000 | 0.0030 | - | - |
| | TAML | 0.5000 | 0.0030 | - | 0.7733 |
| | TA-MAML* | 0.0261 | 0.0005 | 0.0015 | - |
| | MetaSGD | 0.5000 | 0.0030 | - | - |
| | TA-MetaSGD* | 0.0944 | 0.0003 | 0.0002 | - |
| | MetaLSTM | - | 0.005 | - | - |
| | MetaLSTM++ | - | 0.0009 | - | - |
| | TA-MetaLSTM++* | - | 0.0012 | 0.0001 | - |
| | ANIL | 0.5000 | 0.0030 | - | - |
| | TA-ANIL* | 0.0261 | 0.0005 | 0.0015 | - |
| 10.1 | MAML | 0.5000 | 0.0030 | - | - |
| | TAML | 0.5000 | 0.0030 | - | 0.4752 |
| | TA-MAML* | 0.0821 | 0.0002 | 0.0006 | - |
| | MetaSGD | 0.5000 | 0.0030 | - | - |
| | TA-MetaSGD* | 0.0512 | 0.0007 | 0.0018 | - |
| | MetaLSTM | - | 0.005 | - | - |
| | MetaLSTM++ | - | 0.0011 | - | - |
| | TA-MetaLSTM++* | - | 0.0018 | 0.0002 | - |
| | ANIL | 0.5000 | 0.0030 | - | - |
| | TA-ANIL* | 0.0821 | 0.0002 | 0.0006 | - |
| 10.5 | MAML | 0.5000 | 0.0030 | - | - |
| | TAML | 0.5000 | 0.0030 | - | 0.2501 |
| | TA-MAML* | 0.0821 | 0.0002 | 0.0006 | - |
| | MetaSGD | 0.5000 | 0.0030 | - | - |
| | TA-MetaSGD* | 0.0512 | 0.0007 | 0.0018 | - |
| | MetaLSTM | - | 0.0050 | - | - |
| | MetaLSTM++ | - | 0.0024 | - | - |
| | TA-MetaLSTM++* | - | 0.0015 | 0.0019 | - |
| | ANIL | 0.5000 | 0.0030 | - | - |
| | TA-ANIL* | 0.0821 | 0.0002 | 0.0006 | - |

