# OpenReview forum: "Not All Tasks are Equal - Task Attended Meta-learning for Few-shot Learning"
_TMLR — Rejected by TMLR_

### Review · Reviewer_8WoW · 2022-12-04

**Summary Of Contributions:**

In this paper the authors propose a training schedule and attention module for weighting the tasks in a meta-batch based on their importance. The authors compare the proposed method on various datasets (e.g. miniImageNet, CF100, tieredImageNet, etc), methods (e.g. MAML, ANIL, MetaSGD), conditions (e.g. cos-domain), and against concurrent sampling approaches (e.g. ATS, GCP). Ablations are presented to highlight the importance of each component provided to the attention module.

Overall the paper has improved from the previous version, and the empirical evidences are now more solid. However, I have some doubts regarding the overall impact of the work that I have discussed below in more detail.

**Audience:**

Yes

**Broader Impact Concerns:**

None.

**Claims And Evidence:**

Yes

**Requested Changes:**

Refer to the points described in the "Weaknesses" section above.

**Strengths And Weaknesses:**

Strengths
---------

- The contribution of a task in the learning dynamics is an important topic that has not received a lot of attention.
- The authors provide a large number of experiments that gives a good overview of the capabilities of the proposed method.
- The paper is overall clear, and has significantly improved from the earlier version.

Weaknesses
----------

1. My main concern regards the novelty and overall impact of the paper. A recent line of work has showed that standard fine-tuning methods (often called "transfer learning" methods) can achieve state-of-the-art accuracy on a variety of visual classification problems (e.g. Big Transfer, Kolesnikov et al., 2020). The performance gains obtained with the proposed method seems marginal when compared with the performances of those state-of-the-art fine-tuners. Note that, other classes of meta-learners (e.g. ProtoNets, CNAPs-variants, etc) have also proved to be able to scale to much more challenging benchmarks with good performance (Bronskill et al., 2021). All these methods are able to scale to large datasets/benchmarks such as MetaDataset (Triantafillou et al. 2019) and VTAB (Zhai et al. 2019), using deep architectures (e.g. ResNet50 or larger), and large images, while this seems more complicated for the proposed approach. My point here is that while the authors show improvements over standard MAML-like approaches, if we zoom out and consider the most recent developments in the filed then those improvements become less significant. It is not clear to me how this paper can be framed when we consider this larger perspective. I would like to see a discussion of the authors about this point.

2. As a corollary of the previous point, while the method has been tested under domain shift (Section 5.4) the nature of the shift is rather limited. The method has been trained on miniImagenet and tested on CUB-200 and FGVC-Aircraft, that are composed of objects from similar classes. Generalization in this setting may be easier. I am not sure that the proposed method would be able to generalize well to very different tasks. Training on miniImagenet (or ImageNet) and testing on VTAB may be necessary to prove this point.


References
----------

Bronskill, J., Massiceti, D., Patacchiola, M., Hofmann, K., Nowozin, S., & Turner, R. (2021). Memory efficient meta-learning with large images. Advances in Neural Information Processing Systems, 34, 24327-24339.

Kolesnikov, A., Beyer, L., Zhai, X., Puigcerver, J., Yung, J., Gelly, S., & Houlsby, N. (2020, August). Big transfer (bit): General visual representation learning. In European conference on computer vision (pp. 491-507). Springer, Cham.

Triantafillou, E., Zhu, T., Dumoulin, V., Lamblin, P., Evci, U., Xu, K., ... & Larochelle, H. (2019). Meta-dataset: A dataset of datasets for learning to learn from few examples. arXiv preprint arXiv:1903.03096.

Zhai, X., Puigcerver, J., Kolesnikov, A., Ruyssen, P., Riquelme, C., Lucic, M., ... & Houlsby, N. (2019). A large-scale study of representation learning with the visual task adaptation benchmark. arXiv preprint arXiv:1910.04867.

---

> ### Author Response · Authors · 2022-12-26
> **Response to Reviewer 8WoW (part 1)**
>
> We are thankful to the reviewer for the constructive comments. The reviewer acknowledges that investigating the contribution of tasks in the learning meta-model is an understudied yet relevant problem. We are grateful for the appreciative comment of the reviewer that the paper is now empirically more convincing and clear.
>
>
> **Concern 1**
>
>
> *My main concern regards the novelty and overall impact of the paper. A recent line of work has showed that standard fine-tuning methods (often called "transfer learning" methods) can achieve state-of-the-art accuracy on a variety of visual classification problems (e.g. Big Transfer, Kolesnikov et al., 2020). The performance gains obtained with the proposed method seems marginal when compared with the performances of those state-of-the-art fine-tuners. Note that, other classes of meta-learners (e.g. ProtoNets, CNAPs-variants, etc) have also proved to be able to scale to much more challenging benchmarks with good performance (Bronskill et al., 2021). All these methods are able to scale to large datasets/benchmarks such as MetaDataset (Triantafillou et al. 2019) and VTAB (Zhai et al. 2019), using deep architectures (e.g. ResNet50 or larger), and large images, while this seems more complicated for the proposed approach. My point here is that while the authors show improvements over standard MAML-like approaches, if we zoom out and consider the most recent developments in the filed then those improvements become less significant. It is not clear to me how this paper can be framed when we consider this larger perspective. I would like to see a discussion of the authors about this point.*
>
> **Answer to Concern 1**
>
> Transfer learning and meta-learning are two approaches that are commonly used to address few-shot learning problems. Transfer learning involves learning generalizable representations from larger datasets and models, and then using simple algorithms like fine-tuning to adapt to the specific task at hand. On the other hand, meta-learning approaches aim to find an algorithmic solution to few-shot learning. Due to their simplicity, transfer learning approaches scale well with larger image sizes and deeper models. In contrast, meta-learning approaches are memory intensive, which has become a barrier in scaling them to larger image sizes and deeper backbones [9]. Addressing the computational issues of meta-learning approaches and scaling them to larger support sets, deeper backbones and larger image sizes is a concurrent area of research [1, 11]. We leave the integration of our approach with these techniques to enhance the scalability to the future.
>
>
> Equipped with deeper backbones and larger image sizes, transfer learning approaches achieved high performances, particularly in cross-domain settings [1, 7, 8, 9]. However, a line of literature [1] suggests meta-learning approaches may be better suited for **constrained test settings**. This is because transfer learning relies on large pre-trained feature extractors and may require hundreds of optimization steps and careful hyperparameter tuning to perform well [1, 2]. For example, Meta-dataset Transfer approach [3] finetunes all parameters of a ResNet18 feature backbone with a cosine classifier head for 200 optimization steps. Similarly, BiT [2] finetunes the feature backbone with a linear head, sometimes up to 20,000 optimization steps, to acquire state-of-the-art performance on VTAB dataset. Further, transfer learning approaches require significant hyper-parameter tuning on validation sets of each downstream task that also adds to the cost. On the other hand, meta-learning approaches can generalize to unseen meta-test tasks with just a few adaptation steps and often with little or no hyperparameter tuning [1]. While transfer learning may be a better choice in some contexts, meta-learning can be a practical option in cases where computational resources are limited or when the task needs to be adapted on the fly. Overall, both approaches have their own strengths and can be useful in different settings.
>
>
> Our work focuses on a resource-constrained setting, where the number of support instances and the computing available for meta-test adaptation are limited. As a result, our study is confined to meta-learning setups. Attention to examples is a well-explored area in high-shot supervised learning [5, 6], but it is a relatively understudied topic in the episodic training regimen. Recent works such as ATS [14] and uniform sampling [15] have focused on scheduling tasks in episodic training. We clearly distinguish our approach from these state-of-the-art schedulers (Section 2) and demonstrate the superiority of our approach through empirical results in Tables 2, 3, 4, and 5. We, therefore, position our work as a continuation of these studies. We now add this discussion as a part of related work.

---

> > ### Author Response · Authors · 2022-12-26
> > **Response to Reviewer 8WoW (part 2)**
> >
> > We also acknowledge that metric approaches like [3, 1] use more advanced backbones and diverse datasets such as VTAB and Metadataset, and our approach is limited to a basic architecture (Conv4) and gradient-based methods. We clarify that though our approach is extendible to any episodic curriculum (including metric approaches with minor design changes), we choose gradient-based approaches like MAML and ANIL approaches as they are domain-agnostic in contrast to metric learning. However, we leave the investigation of attention mechanisms for metric approaches and domains, such as reinforcement learning or regression problems for gradient approaches for future work.
> >
> > We also like to direct the reviewer's attention to the existing literature [10] that compares vanilla transfer learning (with no Imaganet pretraining or data augmentation) for conv4 backbone with episodic training (MAML) under fair conditions. Chen et al. have demonstrated that MAML performs better than vanilla transfer learning under fair conditions for conv4 architecture. However, transfer learning scales much better with the architectures than MAML (or other episodic methods) [10]. Nevertheless, transfer learning (TL) is a good solution for few-shot learning (especially with Imagenet pretraining and larger backbones), and translating attention to TL for a few-shot setup is a promising direction for further research.
> >
> > In the next section, we have addressed the reviewer's primary concern and evaluated the proposed approach on the suggested diverse VTAB dataset. We acknowledge that similar to [12, 13, 14] our study is limited to understanding the fundamentals of episodic training rather than developing an algorithm that surpasses the state-of-the-art approach for few shot learning. We add this discussion as a broader impact section in the revised draft.
> >
> > **Concern 2:**
> >
> > *As a corollary of the previous point, while the method has been tested under domain shift (Section 5.4) the nature of the shift is rather limited. The method has been trained on miniImagenet and tested on CUB-200 and FGVC-Aircraft, that are composed of objects from similar classes. Generalization in this setting may be easier. I am not sure that the proposed method would be able to generalize well to very different tasks. Training on miniImagenet (or ImageNet) and testing on VTAB may be necessary to prove this point.*
> >
> > **Response to concern 2:**
> >
> > We agree with the reviewer that some classes overlap between the Imagenet and CUB-200, and FGVC-Aircraft datasets. Thus, training the model on miniImagenet and evaluating it on CUB or Aircraft dataset is not fully representative of the cross-domain setup. Upon the reviewer's suggestion, we now perform an additional set of experiments on the diverse VTAB dataset, which does not share classes with the Imagenet (consequently miniImagenet) dataset. We note that some VTAB sub-datasets like Sun397 are quite memory intensive and others like Patch Camelyon, Retinopathy, etc., have fewer classes. **In the interest of time and with the available resources, we meta-train a conv4 model on the miniImagenet dataset** and evaluate it on a few of feasible sub-datasets covering all three domains - Natural, Specialized, and Structured. Specifically, we investigate the merit of the proposed approach on Natural sub-datasets like DTD, CIFAR FC 100, Flowers102, and SVHN, specialized sub-datasets like EuroSAT and Resisc45, and structured sub-datasets like dSprites/location and dSprites/orientation. We convert the selected VTAB sub-datasets to a few-shot setup (5-way 1 and 5 shot tasks) and evaluate task-attended MAML (TA-MAML) and its vanilla version (MAML) on 300 tasks. Our experiments (Table presented below) demonstrate that task attention allows MAML to better generalize to unseen, diverse out-of-distribution VTAB meta-test sets. We added this experiment to section 5.4 in the revised draft.

---

> > > ### Author Response · Authors · 2022-12-26
> > > **Response to Reviewer 8WoW (part 3)**
> > >
> > > Table 5: Comparative analysis of proposed approach (TA-MAML) and its non-task attended counterpart (MAML) after training on the miniImagenet dataset and tested on VTAB (Natural: CIFAR FC 100, Flowers102, and SVHN, Specialized: EuroSAT and Resisc45, Structured: dSprites/location and dSprites/orientation datasets) for 5 way 1 and 5 shot settings.
> > >
> > >
> > > | Algorithm  |                   |  Test Accuracy |                      |                |
> > > |------------|:-----------------:|:--------------:|:--------------------:|:--------------:|
> > > |            | 5 way 1 shot      | 5 way 5 shot   | 5 way 1 shot         | 5 way 5 shot   |
> > > |            |  **Flowers102**  |    | **FC100**               |                |
> > > | MAML#      | 51.93 ± 1.59      | 75.22 ± 0.48   | 35.49 ± 1.95         | 44.42 ± 0.83   |
> > > | **TA-MAML#**   | **61.86 ± 1.72**      | **77.49 ± 0.16**   | **38.87 ± 1.90**         | **46.57 ± 0.85**   |
> > > |            |        **SVHN**       |                |                      |                |
> > > | MAML#      | 20.93 ± 1.01      | 22.42 ± 0.88   |                      |                |
> > > | **TA-MAML#**   | **21.73 ± 1.09**     | **24.20 ± 0.78**   |                      |                |
> > > |            |      **EuroSAT**     |                |        **Resisc45**      |                |
> > > | MAML#      | 45.80 ± 1.49      |  62.0 ± 0.71   | 33.60 ± 1.49         |  42.07 ± 0.37  |
> > > | **TA-MAML#**   | **51.67 ± 1.62**      |  **66.69 ± 0.70**  | **35.20 ± 1.21**         |  **46.27 ± 0.39**  |
> > > |            | **DSprites_location** |                | **DSprites_orientation** |                |
> > > | MAML#      | 36.67 ± 1.55      | 48.91 ± 0.84   |  20.86 ± 1.81        | 22.89 ± 0.95   |
> > > | **TA-MAML#**   | **39.93 ± 1.33**      | **56.48 ± 0.95**   | **24.27 ± 1.18**         | **22.92 ± 0.93**   |
> > >
> > > ---------------------
> > > ---------------------
> > > **References**
> > >
> > > [1] Bronskill, J., Massiceti, D., Patacchiola, M., Hofmann, K., Nowozin, S., & Turner, R. (2021). Memory efficient meta-learning with large images. Advances in Neural Information Processing Systems, 34, 24327-24339.
> > >
> > > [2] Kolesnikov, A., Beyer, L., Zhai, X., Puigcerver, J., Yung, J., Gelly, S., & Houlsby, N. (2020, August). Big transfer (bit): General visual representation learning. In European conference on computer vision (pp. 491-507). Springer, Cham.
> > >
> > > [3] Triantafillou, E., Zhu, T., Dumoulin, V., Lamblin, P., Evci, U., Xu, K., ... & Larochelle, H. (2019). Meta-dataset: A dataset of datasets for learning to learn from few examples. arXiv preprint arXiv:1903.03096.
> > >
> > > [4] Zhai, X., Puigcerver, J., Kolesnikov, A., Ruyssen, P., Riquelme, C., Lucic, M., ... & Houlsby, N. (2019). A large-scale study of representation learning with the visual task adaptation benchmark. arXiv preprint arXiv:1910.04867.
> > >
> > > [5] Shu, J., Xie, Q., Yi, L., Zhao, Q., Zhou, S., Xu, Z., & Meng, D. (2019). Meta-weight-net: Learning an explicit mapping for sample weighting. Advances in neural information processing systems, 32.
> > >
> > > [6] Li, Z., Wu, Y., Chen, K., Wu, Y., Zhou, S., Liu, J., & Yan, J. (2020, April). Learning to Auto Weight: Entirely Data-driven and Highly Efficient Weighting Framework. In Proceedings of the AAAI Conference on Artificial Intelligence (Vol. 34, No. 04, pp. 4788-4795).
> > >
> > > [7] Guo, Y., Codella, N. C., Karlinsky, L., Codella, J. V., Smith, J. R., Saenko, K., ... & Feris, R. (2020, August). A broader study of cross-domain few-shot learning. In European conference on computer vision (pp. 124-141). Springer, Cham.
> > >
> > > [8] Dhillon, G. S., Chaudhari, P., Ravichandran, A., & Soatto, S. (2019, September). A Baseline for Few-Shot Image Classification. In International Conference on Learning Representations.
> > >
> > > [9] Dumoulin, V., Houlsby, N., Evci, U., Zhai, X., Goroshin, R., Gelly, S., & Larochelle, H. (2021, June). A unified few-shot classification benchmark to compare transfer and meta learning approaches. In Thirty-fifth Conference on Neural Information Processing Systems Datasets and Benchmarks Track (Round 1).
> > >
> > > [10] Chen, W. Y., Liu, Y. C., Kira, Z., Wang, Y. C. F., & Huang, J. B. (2019). A Closer Look at Few-shot Classification. In International Conference on Learning Representations.
> > >
> > > [11] Shin, J., Lee, H. B., Gong, B., & Hwang, S. J. (2021, July). Large-Scale Meta-Learning with Continual Trajectory Shifting. In International Conference on Machine Learning (pp. 9603-9613). PMLR.
> > >
> > > [12] Wu, Y., Huang, L. K., & Wei, Y. Adversarial Task Up-sampling for Meta-learning (2022). In Advances in Neural Information Processing Systems.
> > >
> > > [13] Raghu et.al. Rapid Learning or Feature Reuse? Towards Understanding the Effectiveness of MAML. In ICLR, 2019.
> > >
> > > [14] Yao et.al. Meta-learning with an Adaptive Task Scheduler. NeurIPS, 2021.
> > >
> > > [15] Arnold et.al. Uniform Sampling over Episode Difficulty. NeurIPS, 2021.

---

### Review · Reviewer_yWta · 2022-12-16

**Summary Of Contributions:**

The paper investigates if per-element importance weights on batch elements can improve performance in a meta-learning.
The paper proposes to add an attention network that assigns different importance weights to batch elements in a meta-learning framework. The attention network is trained end-to-end with the meta-learner and uses various hand-crafted features, such as the norm of the gradient, and the ration of losses before/after learning to derive these weights. The proposed method can be combined with various meta-learners and was evaluated with MAML, MetaSGD, metaLSTM, and ANIL.

**Audience:**

Yes

**Broader Impact Concerns:**

None.

**Claims And Evidence:**

Yes

**Requested Changes:**

Please explain the gap in performance with SUR and why meta-learning is still a promising approach to few-shot learning.

**Strengths And Weaknesses:**

Strengths:
Considerable evaluations and ablation studies. I appreciate the author's efforts to clarify the difference in performance from published/reproduced MAML for example, as well as parameter tuning details.
Intuitively convincing ablation/analyses. The authors show that attention weights do not have a clear interpretation in all cases. Also I like the nice plot in Figure 5, which reinforces the intuition that all samples are equally important early in training but certain samples are revealed to be more "difficult" only later in training.

Weaknesses:
Lack of evaluation on MetaDataset, one of the largest dataset for meta-learning evaluation.
The performance seems to be far below recent works, which are far simpler algorithmically.
For example "Selecting Relevant Features from a Multi-domain Representation for Few-shot Classification (SUR)" by Dvornik et al. 2020 achieves 81.19 ± 0.41%, 63.93 ± 0.63% on 5-way 5-shot and 1-shot mini-ImageNet, compared to the author's best performing model TA-MAML 65.69 ± 1.08%, 50.35 ± 0.1.22%. SUR uses no meta-learning at all, simply supervised training with nearest neighbor-like classification.

---

> ### Author Response · Authors · 2022-12-26
> **Response to Reviewer yWta (part 1)**
>
> We are grateful to the reviewer for the constructive comments. We thank the reviewer for acknowledging the quantitative and qualitative analysis conducted in the paper. We address the reviewer's concerns as follows:
>
>
> **Concern1**
>
>
> *Why is meta-learning still a promising approach to few-shot learning?*
>
> **Response to concern1**
>
>
> Transfer learning and meta-learning are two approaches that are commonly used to address few-shot learning problems. Transfer learning involves learning generalizable representations from larger datasets and models, and then using simple algorithms like fine-tuning to adapt to the specific task at hand. On the other hand, meta-learning approaches aim to find an algorithmic solution to few-shot learning. Due to their simplicity, transfer learning approaches scale well with larger image sizes and deeper models. In contrast, meta-learning approaches are memory intensive, which has become a barrier in scaling them to larger image sizes and deeper backbones [9]. Addressing the computational issues of meta-learning approaches and scaling them to larger support sets, deeper backbones and larger image sizes is a concurrent area of research [1, 11]. We leave the integration of our approach with these techniques to enhance the scalability to the future.
>
> Equipped with deeper backbones and larger image sizes, transfer learning approaches achieved high performances, particularly in cross-domain settings [1, 7, 8, 9]. However, a line of literature [1] suggests meta-learning approaches may be better suited for **constrained test settings**. This is because transfer learning relies on large pre-trained feature extractors and may require hundreds of optimization steps and careful hyperparameter tuning to perform well [1, 2]. For example, Meta-dataset Transfer approach [3] finetunes all parameters of a ResNet18 feature backbone with a cosine classifier head for 200 optimization steps. Similarly, BiT [2] finetunes the feature backbone with a linear head, sometimes up to 20,000 optimization steps, to acquire state-of-the-art performance on VTAB dataset. Further, transfer learning approaches require significant hyper-parameter tuning on validation sets of each downstream task that also adds to the cost. On the other hand, meta-learning approaches can generalize to unseen meta-test tasks with just a few adaptation steps and often with little or no hyperparameter tuning [1]. While transfer learning may be a better choice in some contexts, meta-learning can be a practical option in cases where computational resources are limited or when the task needs to be adapted on the fly. Overall, both approaches have their own strengths and can be useful in different settings. We now add this discussion as a part of related work.
>
> **Concern 2**
>
>
> *Explain the gap in performance with SUR.*
>
> **Response to concern 2**
>
> Our approach uses Conv4 for its backbone; thus, the reported results are not comparable with SUR which uses Resnet-12 for miniImagenet and Resnet-18 for meta-dataset. SUR also includes pretraining on Imagenet, data augmentation, learning rate schedule and other techniques; while we use a standard off-the-shelf training setup. Unfortunately, we lack the necessary resources to experiment with deeper backbones, so, similar to [12, 13, 14], we limit our study to understanding the fundamentals of episodic training and leverage unequal contribution of tasks instead of attempting to surpass the state-of-the-art approach for few shot learning.
>
> SUR is a cost-effective way to adapt a backbone at test time, and helps to reduce negative transfer in cross-domain settings. It is different from our approach that is designed to study and leverage the unequal contribution of tasks to learn a meta-model during training. Additionally, our approach can be expanded to algorithms that only adjust the last layer during testing (TA-ANIL) and even to non-adaptive algorithms such as prototypical networks with slight modifications. Training our approach requires an extra 4-layer attention network in addition to the base model, yet it is less computationally expensive than SUR which needs to train various base models (e.g. 7 parallel Resent-18 backbones for Metadataset) or FiLM layers, depending on the number of training datasets.

---

> > ### Author Response · Authors · 2022-12-26
> > **Response to Reviewer yWta (part 2)**
> >
> > **Concern 3**
> >
> > *Lack of evaluation on MetaDataset*
> >
> >
> > **Response to concern 3**
> >
> > We conduct experiments to evaluate the performance of our proposed approach in cross domain settings. We trained our model episodically on miniImagenet and evaluated it on two datasets - CUB200 and FGVC aircrafts - from Metadataset, and the results showed that our approach is superior to the state of the art uniform sampling approach (Table 5 main paper). Additionally, at the reviewers' suggestion, we now conduct further experiments on Describable features and Omniglot datasets from diverse Metadataset to verify the merit of our task-attended approach over state-of-the art uniform sampling approach (wherever applicable) or non-task-attended counterparts. We add these results to section 5.4 of the main paper.
> >
> >
> > | Algorithm                |   |             |         Test Accuracy on Metadatset        |              |
> > |--------------------------|:-------------------:|:---------------------------:|:---------------------------:|:---------------------------:|
> > || 5 way 1 shot                | 5 way 5 shot            | 5 way 1 shot         | 5 way 5 shot|
> > ||           **CUB 200**           |               | **FGVC-Aircraft**||
> > | MAML+ UNIFORM (Online) # | 35.84 ± 0.54| 46.67 ± 0.55  | 26.62 ± 0.39  | 34.41 ± 0.44 |
> > | **TA-MAML#**| **42.87 ± 0.98**| **57.49 ± 0.79**      | **29.42 ± 0.78**  | **36.34 ± 0.86** |
> > ||     **Describable Textures**    ||||
> > | MAML+ UNIFORM (Online)#  | 31.84 ± 0.49| 40.81 ± 0.44  |||
> > | **TA-MAML#**| **31.98 ± 0.98**| **44.39 ± 0.79**  |||
> > ||**Omniglot**          ||||
> > | MAML#| 72.40 ± 1.43| 86.81 ± 0.99  |||
> > | **TA-MAML#**| **78.73 ± 1.08**| **88.92 ± 0.76**  |||
> >
> > ------------
> > ------------
> >
> > **References:**
> >
> > [1] Bronskill, J., Massiceti, D., Patacchiola, M., Hofmann, K., Nowozin, S., & Turner, R. (2021). Memory efficient meta-learning with large images. Advances in Neural Information Processing Systems, 34, 24327-24339.
> >
> > [2] Kolesnikov, A., Beyer, L., Zhai, X., Puigcerver, J., Yung, J., Gelly, S., & Houlsby, N. (2020, August). Big transfer (bit): General visual representation learning. In European conference on computer vision (pp. 491-507). Springer, Cham.
> >
> > [3] Triantafillou, E., Zhu, T., Dumoulin, V., Lamblin, P., Evci, U., Xu, K., ... & Larochelle, H. (2019). Meta-dataset: A dataset of datasets for learning to learn from few examples. arXiv preprint arXiv:1903.03096.
> >
> > [4] Zhai, X., Puigcerver, J., Kolesnikov, A., Ruyssen, P., Riquelme, C., Lucic, M., ... & Houlsby, N. (2019). A large-scale study of representation learning with the visual task adaptation benchmark. arXiv preprint arXiv:1910.04867.
> >
> > [5] Shu, J., Xie, Q., Yi, L., Zhao, Q., Zhou, S., Xu, Z., & Meng, D. (2019). Meta-weight-net: Learning an explicit mapping for sample weighting. Advances in neural information processing systems, 32.
> >
> > [6] Li, Z., Wu, Y., Chen, K., Wu, Y., Zhou, S., Liu, J., & Yan, J. (2020, April). Learning to Auto Weight: Entirely Data-driven and Highly Efficient Weighting Framework. In Proceedings of the AAAI Conference on Artificial Intelligence (Vol. 34, No. 04, pp. 4788-4795).
> >
> > [7] Guo, Y., Codella, N. C., Karlinsky, L., Codella, J. V., Smith, J. R., Saenko, K., ... & Feris, R. (2020, August). A broader study of cross-domain few-shot learning. In European conference on computer vision (pp. 124-141). Springer, Cham.
> >
> > [8] Dhillon, G. S., Chaudhari, P., Ravichandran, A., & Soatto, S. (2019, September). A Baseline for Few-Shot Image Classification. In International Conference on Learning Representations.
> >
> > [9] Dumoulin, V., Houlsby, N., Evci, U., Zhai, X., Goroshin, R., Gelly, S., & Larochelle, H. (2021, June). A unified few-shot classification benchmark to compare transfer and meta learning approaches. In Thirty-fifth Conference on Neural Information Processing Systems Datasets and Benchmarks Track (Round 1).
> >
> > [10] Chen, W. Y., Liu, Y. C., Kira, Z., Wang, Y. C. F., & Huang, J. B. (2019). A Closer Look at Few-shot Classification. In International Conference on Learning Representations.
> >
> > [11] Shin, J., Lee, H. B., Gong, B., & Hwang, S. J. (2021, July). Large-Scale Meta-Learning with Continual Trajectory Shifting. In International Conference on Machine Learning (pp. 9603-9613). PMLR.
> >
> > [12] Wu, Y., Huang, L. K., & Wei, Y. Adversarial Task Up-sampling for Meta-learning (2022). In Advances in Neural Information Processing Systems.
> >
> > [13] Raghu et.al. Rapid Learning or Feature Reuse? Towards Understanding the Effectiveness of MAML. In ICLR, 2019.
> >
> > [14] Yao et.al. Meta-learning with an Adaptive Task Scheduler. NeurIPS, 2021.
> >
> > [15] Arnold et.al. Uniform Sampling over Episode Difficulty. NeurIPS, 2021.

---

### Review · Reviewer_uLbm · 2022-12-31

**Summary Of Contributions:**

In this paper, the authors investigate the importance of each task sampled via episodic sampling while training a model using meta-learning. Towards this, they propose to use an auxiliary network, which based on certain features from each task (norm-of-gradients, val-acc, val-loss etc) learns a relative weighting for each task in a segment and uses that weighting to adjust the contribution of each task in the meta-learning training loss. Authors show through a variety of (relatively small-scale) experiments that adding such a learned task importance module in the meta-learning training paradigm can improve the performance of the model.

**Audience:**

Yes

**Broader Impact Concerns:**

N/A.

**Claims And Evidence:**

Yes

**Requested Changes:**

Please see my comments regarding weaknesses.

* The proposed method needs to be explained better; the diagram can be made more intuitive and the whole workflow e.g. how the tasks are getting sampled and being fed into the end-to-end learner should be explained in a more succinct manner.
* Experimental results need to be performed using networks which are used in practice and should compare against SOTA methods on the benchmark tasks. For example, [this paper](https://arxiv.org/pdf/1911.04623.pdf) from 2019 achieves 64/81 1-shot/5-shot performance on Mini-ImageNet whereas this paper reports 53/66 (MAML+TA, Table 3).
* [Minor] Connecting the proposed method to existing methods from ML/NLP literature on attention will help the reader better understand where the attention part is coming from.


**Strengths And Weaknesses:**

Strengths
------------
* Overall, I like the idea when it comes to learning the importance of each task using certain characteristics from the task (which are properly ablated in the paper) and incorporating that weighting in the end-to-end meta-learning procedure. I also agree with the claim of the paper and associated experiments that this proposed technique is generic and can be added to any meta-learning algorithm that uses episodic sampling.
* I like the breadth of the experiments performed in terms of core results, cross-domain few-shot results, comparing against uniform sampling and the other paper, ablation studies etc.

Weaknesses
---------------
* As (probably) suggested by the previous reviewer, the experiments are done with a toy network and primarily compares against MAML, which is a method from 2017 and is quite far when compared to state-of-the-art few-shot learning methods. For cross-domain, the paper only evaluates the method on a selected set of datasets. In order to truly understand the efficacy of the proposed method, the work needs to be applied to a reasonably sized network (e.g. ResNet-18/34 or ViT-B/32) and compared against state-of-the-art few-shot learning methods on Mini/Tiered-ImageNet and on the full Meta-Dataset test-suite. Outperforming MAML or ANIL is not sufficient at this point.

* The actual method is not well explained e.g. I don't understand how the task weights are computed and then plugged back into the meta-learning procedure. Typically, in an episodic training regime, we randomly sample one task, meta-train using it, then sample another task, meta-train with that and so on. But in this case, authors need to have a set of tasks first to compute the relative importance over those and then only those tasks can be used in the meta-training process. I am not clear about the specifics of how this implemented - are you sampling N tasks first, computing the importance score, meta-learning using it and then again picking N new tasks? If so, how large is N in your implementation? Apologies if this is mentioned in the paper and I missed it.

* I do not understand why the task importance module is dubbed as task attention. It seems to me like a feed-forward network which is taking a set of features per task and applying a softmax layer on the output. Is there any attention between the tasks (e.g. self attention) or some other form of attention (e.g. cross-attention) being at play here?

---

> ### Author Response · Authors · 2023-01-08
> **Response to Reviewer uLbm (Part 1)**
>
> We have addressed the reviewers' concerns below, and all changes made to the draft are marked in blue.
>
>
> **Weakness 1:**
>
> *As (probably) suggested by the previous reviewer, the experiments are done with a toy network and primarily compares against MAML, which is a method from 2017 and is quite far when compared to state-of-the-art few-shot learning methods. For cross-domain, the paper only evaluates the method on a selected set of datasets. In order to truly understand the efficacy of the proposed method, the work needs to be applied to a reasonably sized network (e.g. ResNet-18/34 or ViT-B/32) and compared against state-of-the-art few-shot learning methods on Mini/Tiered-ImageNet and on the full Meta-Dataset test-suite. Outperforming MAML or ANIL is not sufficient at this point.*
>
> **Response to Weakness 1:**
>
> We clarify that we aim to understand and utilize the different contributions of tasks in a batch rather than creating a cutting-edge few-shot learning algorithm. We agree with the reviewer that we have augmented the proposed approach with algorithms published from 2017-2019 (MAML -ICML 2017, MetaSGD -arxiv 2017, MetaLSTM - ICLR 2017, and ANIL - ICLR 2019). We, however, note that this was done to 1) demonstrate our approach can be integrated with various episodic curriculums 2) as the state-of-the-art meta-learning scheduling algorithms (ATS - NeurIPS -2021, Uniform Sampling - NeurIPS -2021) were built on these fundamental algorithms, integration with them was essential to achieve a fair comparison. We emphasize that our approach outperformed multiple task scheduling approaches, such as Focal loss [6] (ICCV 2017), DAML [5] (MICCAI, 2020), PAML [3] (NeurIPS 2020), GCP [4] (ECCV 2020) as well as state-of-the-art task schedulers such as ATS [1] (NeurIPS 2021) and uniform sampling approach [2] (NeurIPS 2021).
>
> We recognize that transfer learning has surpassed meta-learning in terms of scalability to deep architectures [11]. However, a wide range of literature (published in 2021-22 in venues such as NeurIPS, ICML, ICLR, and AAAI) examines new directions or improving existing meta-learning concepts [7,8,9,10,12], all of which build on foundational meta-learning algorithms (like MAML, iMAML, Reptile, MetaSGD, and ANIL) on the conv4 backbone [7,8,9,10,12]. We believe that both research directions, which aim to obtain state-of-the-art performances on few-shot learning [15,16,17,18,19,20] as well as those that tackle specific problems in meta-learning [7,8,9,10,12,14], are of great importance. **We place our approach in the second category**.
>
>
> We also note that the central focus of the paper is not on cross-domain few-shot learning. Upon the reviewer's suggestions, we add the results of a model (conv4) meta-trained on the miniImagenet dataset, evaluated on CUB 200, FGVC-Aircraft, Describable features, and Omniglot datasets from Metadataset (Table presented below) to the section 5.4 of the main paper. We verify the advantage of our task-attended approach over the state-of-the-art uniform sampling approach (NeurIPS 2021) or non-task-attended counterparts on the Metadataset.
>
>
> | Algorithm                |   |             |         Test Accuracy on Metadatset        |              |
> |--------------------------|:-------------------:|:---------------------------:|:---------------------------:|:---------------------------:|
> || 5 way 1 shot                | 5 way 5 shot            | 5 way 1 shot         | 5 way 5 shot|
> ||           **CUB 200**           |               | **FGVC-Aircraft**||
> | MAML+ UNIFORM (Online) # | 35.84 ± 0.54| 46.67 ± 0.55  | 26.62 ± 0.39  | 34.41 ± 0.44 |
> | **TA-MAML#**| **42.87 ± 0.98**| **57.49 ± 0.79**      | **29.42 ± 0.78**  | **36.34 ± 0.86** |
> ||     **Describable Textures**    ||||
> | MAML+ UNIFORM (Online)#  | 31.84 ± 0.49| 40.81 ± 0.44  |||
> | **TA-MAML#**| **31.98 ± 0.98**| **44.39 ± 0.79**  |||
> ||**Omniglot**          ||||
> | MAML#| 72.40 ± 1.43| 86.81 ± 0.99  |||
> | **TA-MAML#**| **78.73 ± 1.08**| **88.92 ± 0.76**  |||

---

> > ### Author Response · Authors · 2023-01-08
> > **Response to Reviewer uLbm (Part 2)**
> >
> > We also conducted an additional set of experiments on the diverse VTAB dataset.  Specifically, we studied the advantages of the proposed approach on Natural datasets- DTD, CIFAR FC 100, Flowers102, SVHN, Specialized datasets - EuroSAT, Resisc45, Structured datasets - dSprites/location and dSprites/orientation for the cross-domain setting. To adhere to [7], we kept Describable Textures as part of Metadataset and Flowers102 as a component of the VTAB dataset. Our experiments (as seen in the Table below) show that task attention allows MAML to better generalize to unseen, distinct out-of-distribution VTAB meta-test sets. This experiment has been added to section 5.4 in the revised draft.
> >
> >
> >
> >
> > Table 5: Comparative analysis of proposed approach (TA-MAML) and its non-task attended counterpart (MAML) after training on the miniImagenet dataset and tested on VTAB (Natural: CIFAR FC 100, Flowers102, and SVHN, Specialized: EuroSAT and Resisc45, Structured: dSprites/location and dSprites/orientation datasets) for 5 way 1 and 5 shot settings.
> >
> >
> > | Algorithm  |                   |  Test Accuracy |                      |                |
> > |------------|:-----------------:|:--------------:|:--------------------:|:--------------:|
> > |            | 5 way 1 shot      | 5 way 5 shot   | 5 way 1 shot         | 5 way 5 shot   |
> > |            |  **Flowers102**  |    | **FC100**               |                |
> > | MAML#      | 51.93 ± 1.59      | 75.22 ± 0.48   | 35.49 ± 1.95         | 44.42 ± 0.83   |
> > | **TA-MAML#**   | **61.86 ± 1.72**      | **77.49 ± 0.16**   | **38.87 ± 1.90**         | **46.57 ± 0.85**   |
> > |            |        **SVHN**       |                |                      |                |
> > | MAML#      | 20.93 ± 1.01      | 22.42 ± 0.88   |                      |                |
> > | **TA-MAML#**   | **21.73 ± 1.09**     | **24.20 ± 0.78**   |                      |                |
> > |            |      **EuroSAT**     |                |        **Resisc45**      |                |
> > | MAML#      | 45.80 ± 1.49      |  62.0 ± 0.71   | 33.60 ± 1.49         |  42.07 ± 0.37  |
> > | **TA-MAML#**   | **51.67 ± 1.62**      |  **66.69 ± 0.70**  | **35.20 ± 1.21**         |  **46.27 ± 0.39**  |
> > |            | **DSprites_location** |                | **DSprites_orientation** |                |
> > | MAML#      | 36.67 ± 1.55      | 48.91 ± 0.84   |  20.86 ± 1.81        | 22.89 ± 0.95   |
> > | **TA-MAML#**   | **39.93 ± 1.33**      | **56.48 ± 0.95**   | **24.27 ± 1.18**         | **22.92 ± 0.93**   |
> >
> >
> > **Weakness 2:**
> >
> > *The actual method is not well explained e.g. I don't understand how the task weights are computed and then plugged back into the meta-learning procedure. Typically, in an episodic training regime, we randomly sample one task, meta-train using it, then sample another task, meta-train with that and so on. But in this case, authors need to have a set of tasks first to compute the relative importance over those and then only those tasks can be used in the meta-training process. I am not clear about the specifics of how this implemented - are you sampling N tasks first, computing the importance score, meta-learning using it and then again picking N new tasks? If so, how large is N in your implementation? Apologies if this is mentioned in the paper and I missed it.*
> >
> > **Response to Weakness 2:**
> >
> > The reviewer is correct in noting that (1) we randomly sample a task batch (of size B), (2) adapt the model on each task, (3) compute meta-information for each task model, (4) estimate weights of tasks, and (5) perform a weighted meta-update (Equation 2) before flushing the computational graphs. This procedure is common to methods such as MAML [22], ANIL [13], MetaLSTM [23], and TAML [24] (with the exception of steps 3, 4, and 5). A detailed explanation of the process can be found in Requested Changes 1. We used a standard batch size of 4, as found in [22, 24] (line 324 of the draft). However, we have also investigated the influence of the batch size in Table 1 of the main paper.
> >
> > **Weakness 3:**
> >
> > *I do not understand why the task importance module is dubbed as task attention. It seems to me like a feed-forward network which is taking a set of features per task and applying a softmax layer on the output. Is there any attention between the tasks (e.g. self attention) or some other form of attention (e.g. cross-attention) being at play here?*
> >
> > **Response to Weakness 3:**
> >
> > The reviewer is correct in understanding that the "task attention module" is a simple feedforward network that takes meta-information of each task in a batch, and dispatches weights accordingly. The focus of the meta-module thus varies for each task in a batch, according to the weights provided by the attention module. This is why the term "task attention" is used, (as attention implies focusing on a discrete aspect of information [21]). This is different from conventional self or cross-attention as seen in machine learning literature.

---

> > > ### Author Response · Authors · 2023-01-08
> > > **Response to Reviewer uLbm (Part 3)**
> > >
> > > **Requested Changes 1:**
> > >
> > > *The proposed method needs to be explained better; the diagram can be made more intuitive and the whole workflow e.g. how the tasks are getting sampled and being fed into the end-to-end learner should be explained in a more succinct manner.*
> > >
> > >
> > > **Response to Requested Changes 1:**
> > >
> > >
> > > As suggested by the reviewer, a new diagram (Figure 7) has been created to further improve the comprehension of the proposed approach. We have added this figure and a related explanation to the appendix. We explain the proposed approach through Figure 1, Figure 7, Algorithm 1, and equations as follows:
> > >
> > >
> > > We first sample a batch of tasks ($B$) from a random pool of data (Figure 7 - Label 1). For each task, the base-model $\phi_i$ is adapted using the support data $D_i$ for $T$ time-steps (line 7 and lines 20-32 in Algorithm 1, Figure 7 - Label 3). Specifically, the adaptation is done using gradient descent on the train loss $L$ for initialization approaches (lines 22-26 in Algorithm 1, Figure 7 - GD), or the current loss and gradients are inputted to the meta-model $\theta$ for optimization approaches, which then outputs the updated base-model parameters (lines 27-31 in Algorithm 1, Figure 7 - PO). The meta-information ($I$) corresponding to each task in the batch is then calculated (Figure 7 - Label: 4), which includes the loss, accuracy, loss-ratio, and gradient norm of adapted models on the query data. This is given as input to the task attention module (Figure 1 - Label: 2, Figure 7 - Label: 5), which outputs the attention vector (line 10 in Algorithm 1, Figure 7- Label: 6). The attention vector and test losses are used to update the meta-model parameters $\theta$ according to equation 2 (line 11 in Algorithm 1, Figure 1 - Label: 4, Figure 7 - Label: 7). A new batch of tasks is then sampled and the base-models are adapted using the updated meta-model (Lines 12-16 in Algorithm 1, Figure 1 - Label: 5). The mean test loss over the adapted base-models is calculated and used to update the parameters of the task attention module $\delta$ according to equation 3.
> > >
> > > **Requested Changes 2:**
> > >
> > > *Experimental results need to be performed using networks which are used in practice and should compare against SOTA methods on the benchmark tasks. For example, this paper from 2019 achieves 64/81 1-shot/5-shot performance on Mini-ImageNet whereas this paper reports 53/66 (MAML+TA, Table 3).*
> > >
> > >
> > > **Response to Requested Changes 2:**
> > >
> > > Our approach focuses on task importance in meta-learning rather than transfer learning. Unlike transfer learning, meta-learning approaches can generalize to unseen meta-test tasks with just a few adaptation steps and minimal hyperparameter tuning [14,18]. While the suggested paper achieves better performance than our approach, it is not a task scheduling approach rather a transfer learning + nearest neighbor classification approach. To evaluate our approach, we compared it to existing state-of-the-art task schedulers such as Focal Loss [6] (ICCV 2017), DAML [5] (MICCAI, 2020), PAML [3] (NeurIPS 2020), GCP [4] (ECCV 2020) and ATS [1] (NeurIPS 2021), as well as the uniform sampling approach [2] (NeurIPS 2021).
> > >
> > > Upon reviewer's suggestion, we have added the results of a model (conv4) meta-trained on the miniImagenet dataset and evaluated on Metadataset and VTAB dataset; these results are presented in weaknesses 1 and section 5.4 of the main paper. However, the central focus of the paper is not on cross-domain few-shot learning.
> > >
> > > **Requested Changes 3:**
> > >
> > > *[Minor] Connecting the proposed method to existing methods from ML/NLP literature on attention will help the reader better understand where the attention part is coming from.*
> > >
> > >
> > > **Response to Requested Changes 3:**
> > >
> > > Please note that there is no relevance between the proposed task attended curriculum and the attention mechanisms in the Machine Learning (ML) or Natural Language Processing (NLP) literature. For your reference, an explanation for the term "task attention" is provided in Weakness 3.
> > >
> > >
> > > ---------
> > > ---------
> > >
> > > **References:**
> > >
> > > [1] Yao et.al. Meta-learning with an Adaptive Task Scheduler. NeurIPS, 2021.
> > >
> > > [2] Arnold et.al. Uniform Sampling over Episode Difficulty. NeurIPS, 2021.
> > >
> > > [3] Kaddour et al. Probabilistic active meta-learning. NeurIPS, 2020.
> > >
> > > [4] Liu et al. Adaptive task sampling for meta-learning. ECCV, 2020.
> > >
> > > [5] Li et al. Difficulty-aware meta-learning for rare disease diagnosis. MICCAI, 2020.
> > >
> > > [6] Lin et al. Focal loss for dense object detection. ICCV  2017.
> > >
> > > [7] Jiang et al. Subspace learning for effective meta-learning. ICML, 2022.
> > >
> > > [8] Flennerhag et al. Bootstrapped Meta-Learning. ICLR, 2021.
> > >
> > > [9] Cioba et al. How to distribute data across tasks for meta-learning?. AAAI, 2022.
> > >
> > > [10] Zucchet et al. A contrastive rule for meta-learning. NeurIPS, 2022
> > >
> > > [11] Chen et al. A Closer Look at Few-shot Classification. ICLR, 2019.
> > >
> > > [12] Wu et al. Adversarial Task Up-sampling for Meta-learning. NeurIPS, 2022.

---

> > > > ### Author Response · Authors · 2023-01-08
> > > > **Response to Reviewer uLbm (Part 4)**
> > > >
> > > > [13] Raghu et.al. Rapid Learning or Feature Reuse? Towards Understanding the Effectiveness of MAML. ICLR, 2019.
> > > >
> > > > [14] Bronskill et.al. Memory efficient meta-learning with large images.  NeurIPS 2021.
> > > >
> > > > [15] Kolesnikov et al. Big transfer (bit): General visual representation learning. ECCV, 2020.
> > > >
> > > > [16] Triantafillou et al. Meta-dataset: A dataset of datasets for learning to learn from few examples. ICLR, 2020.
> > > >
> > > > [17] Zhai et al. A large-scale study of representation learning with the visual task adaptation benchmark. arXiv preprint arXiv:1910.04867, 2019.
> > > >
> > > > [18] Dumoulin et al. A unified few-shot classification benchmark to compare transfer and meta learning approaches. NeurIPS, 2021.
> > > >
> > > > [19] Dvornik et al.  Selecting relevant features from a multi-domain representation for few-shot classification. ECCV, 2020.
> > > >
> > > > [20] Wang et al. Simpleshot: Revisiting nearest-neighbor classification for few-shot learning. arXiv, 2019.
> > > >
> > > > [21] https://en.wikipedia.org/wiki/Attention
> > > >
> > > > [22] Finn et.al. Model-agnostic meta-learning for fast adaptation of deep networks. In ICML, 2017.
> > > >
> > > > [23] Ravi et.al. Optimization as a model for few-shot learning. In ICLR, 2017.
> > > >
> > > > [24] Jamal et.al. Task agnostic meta-learning for few-shot learning. In CVPR 2019

---

### Author Response · Authors · 2023-01-06
**General Response**

We kindly request the reviewers to consider our submission through the lens of meta-learning. To this end, we would like to draw your attention to the following points:

1) Our focus is on meta-learning, which is especially suitable for resource-constrained setups, where the number of images per class and the number of adaptation steps in meta-test tasks are limited [14,18].

2) We acknowledge that transfer learning has significantly surpassed meta-learning owing to better scalability to deep architectures [11]. Improving the scalability of meta-learning approaches is a fairly recent development (NeurIPS 2021) [14], and the community is uncertain if the meta-learning approaches (after resolving scalability problems) may regain popularity.

3) A wide range of literature (published in 2021-22 in venues such as NeurIPS, ICML, ICLR, and AAAI) focuses on exploring new directions or unraveling and improvising fundamental concepts of meta-learning [7,8,9,10,12]. These works still build on foundational meta-learning algorithms (like MAML, iMAML, Reptile, MetaSGD, and ANIL) on the conv4 backbone [7,8,9,10,12]. We believe both research directions, which aim to obtain state-of-the-art performances on few shot learning [15,16,17,18,19,20] as well as ones that tackle specific problems in meta-learning [7,8,9,10,12,14], are important.

4) Our goal is not to create a cutting-edge few-shot learning algorithm but to understand and use the varying contributions of tasks in a batch. We agree that we have augmented the proposed approach with algorithms published from 2017-2019 (MAML -ICML 2017, MetaSGD -arxiv 2017, MetaLSTM - ICLR 2017, and ANIL - ICLR 2019). We, however, note that these experiments are intended to show - a) our approach can be integrated with various episodic curriculums b) As the state-of-the-art meta-learning scheduling algorithms (ATS - NeurIPS -2021, Uniform Sampling - NeurIPS -2021) were built on these fundamental algorithms, integration with them was a must for a fair comparison. We clarify that our approach outperformed multiple task scheduling approaches like Focal loss [6] (ICCV 2017), DAML [5] (MICCAI, 2020), PAML [3] (NeurIPS 2020), GCP [4] (ECCV 2020) as well as state-of-the-art task schedulers such as ATS [1] (NeurIPS 2021) and uniform sampling approach [2] (NeurIPS 2021).


-----------------
References:
 ---------------
[1] Yao et.al. Meta-learning with an Adaptive Task Scheduler. NeurIPS, 2021.

[2] Arnold et.al. Uniform Sampling over Episode Difficulty. NeurIPS, 2021.

[3] Kaddour et al. Probabilistic active meta-learning. NeurIPS, 2020.

[4] Liu et al. Adaptive task sampling for meta-learning. ECCV, 2020.

[5] Li et al. Difficulty-aware meta-learning for rare disease diagnosis. MICCAI, 2020.

[6] Lin et al. Focal loss for dense object detection. ICCV  2017.

[7] Jiang et al. Subspace learning for effective meta-learning. ICML, 2022.

[8] Flennerhag et al. Bootstrapped Meta-Learning. ICLR, 2021.

[9] Cioba et al. How to distribute data across tasks for meta-learning?. AAAI, 2022.

[10] Zucchet et al. A contrastive rule for meta-learning. NeurIPS, 2022

[11] Chen et al. A Closer Look at Few-shot Classification. ICLR, 2019.

[12] Wu et al. Adversarial Task Up-sampling for Meta-learning. NeurIPS, 2022.

[13] Raghu et.al. Rapid Learning or Feature Reuse? Towards Understanding the Effectiveness of MAML. ICLR, 2019.

[14] Bronskill et.al. Memory efficient meta-learning with large images.  NeurIPS 2021.

[15] Kolesnikov et al. Big transfer (bit): General visual representation learning. ECCV, 2020.

[16] Triantafillou et al. Meta-dataset: A dataset of datasets for learning to learn from few examples. ICLR, 2020.

[17] Zhai et al. A large-scale study of representation learning with the visual task adaptation benchmark. arXiv preprint arXiv:1910.04867, 2019.

[18] Dumoulin et al. A unified few-shot classification benchmark to compare transfer and meta learning approaches. NeurIPS, 2021.

[19] Dvornik et al.  Selecting relevant features from a multi-domain representation for few-shot classification. ECCV, 2020.

[20] Wang et al. Simpleshot: Revisiting nearest-neighbor classification for few-shot learning. arXiv, 2019.

---

### Decision · Action_Editors · 2023-02-14

**Recommendation:** Reject

**Comment:**

An early version of this paper was rejected with encouragement to resubmit. This is the resubmitted version. The paper was reviewed by three reviewers (who are not reviewers of the early version). Unfortunately all three reviewers recommended "leaning reject" after reviewing the rebuttal.

Reviewer 8WoW thinks that the paper did not shed enough light on the underlying importance of the various tasks in the batch, and the paper does not seem to provide enough empirical evidences on the performance, flexibility, and scalability of the method.

Reviewer yWta thinks that the experiments are in a regime which is largely irrelevant today: small scale networks, outdated meta-learning algorithms (e.g. MAML).

Reviewer uLbm thinks that the proposed algorithm needs to either have enough technical novelty/insight or needs to do better compared to existing non-meta-learning few-shot classification methods.

I must say that I appreciate the authors' insight and efforts. But I also think the reviewers' criticisms are valid. Given the overall negative reviews, I am afraid that I have to recommend reject.


**Audience:**

The proposed method should be interesting to researchers working on meta-learning and few-shot learning.

**Claims And Evidence:**

This paper proposes a meta-learning method where the tasks in a batch are weighted by a learnable task attention module. The effectiveness of the proposed method is illustrated on various datasets.